# Identification and Structural Analysis of Spirostanol Saponin from *Yucca schidigera* by Integrating Silica Gel Column Chromatography and Liquid Chromatography/Mass Spectrometry Analysis

**DOI:** 10.3390/molecules25173848

**Published:** 2020-08-24

**Authors:** Jingya Ruan, Lu Qu, Wei Zhao, Chang Gao, Peijian Huang, Dandan Zheng, Lifeng Han, Haiyang Yu, Zixin Zhang, Yi Zhang, Tao Wang

**Affiliations:** 1Tianjin Key Laboratory of TCM Chemistry and Analysis, Tianjin University of Traditional Chinese Medicine, 10 Poyanghu Road, West Area, Tuanbo New Town, Jinghai District, Tianjin 301617, China; Ruanjy19930919@163.com (J.R.); qululuhan88@163.com (L.Q.); zhaowei126123@126.com (W.Z.); GC087159@163.com (C.G.); hpjforever@sina.com (P.H.); 2Institute of TCM, Yunnan University of Traditional Chinese Medicine, 1076 Yuhua Road, Chenggong District, Kunming 650500, China; 3Institute of TCM, Tianjin University of Traditional Chinese Medicine, 10 Poyanghu Road, West Area, Tuanbo New Town, Jinghai District, Tianjin 301617, China; zhengdd1027@163.com (D.Z.); hanlifeng_1@sohu.com (L.H.); hyyu@tjutcm.edu.cn (H.Y.); zixinzhang0115@163.com (Z.Z.)

**Keywords:** *Yucca schidigera Roezl* (Mojave), spirostanol saponins, chemical constituents profiling, multi-phase liquid chromatography, normal phase chromatography silica column separation, reversed phase liquid chromatography/mass spectrometry analysis

## Abstract

*Yucca schidigera* Roezl (Mojave), a kind of ornamental plant belonging to the *Yucca* genus (Agavaceae), whose extract exhibits important roles in food, beverage, cosmetic and feed additives owing to its rich spirostanol saponins. To provide a comprehensive chemical profiling of the spirostanol saponins in it, this study was performed by using a multi-phase liquid chromatography method combining a reversed phase chromatography T_3_ column with a normal phase chromatography silica column for the separation and an ESI-Q-Exactive-Orbitrap MS in positive ion mode as the detector. By comparing the retention time and ion fragments with standards, thirty-one spirostanol saponins were identified. In addition, according to the summary of the chromatographic retention behaviors and the MS/MS cleavage patterns and biosynthetic pathway, another seventy-nine spirostanol saponins were speculatively identified, forty ones of which were potentially new ones. Moreover, ten novel spirostanol saponins (three pairs of (25*R/S*)-spirostanol saponin isomer mixtures) were targeted for isolation to verify the speculation. Then, the comprehensive chemical profiling of spirostanol saponins from *Y. schidigera* was reported here firstly.

## 1. Introduction

*Yucca schidigera* Roezl (Mojave), belonging to the genus *Yucca*, Agavaceae family, is mainly distributed in the southwestern United States and the northern deserts of Mexico [1]. It is a commonly used commercial raw material for steroid saponins, which mainly contains spirostanol, isospirostanol and furostanol saponins. Modern pharmacological studies have shown that the extract of *Y. schidigera* possessed many bioactivities, such as regulating energy metabolism and improving animal breeding [2,3]. Therefore, it has been widely used in foods, feeds, and cosmetic additives [4]. According to the references, pharmacological studies were mainly focused on its extracts, while the identity of its effective substance(s) is not clear yet since the lack of systematic phytochemistry studies. Furthermore, the lack of quality assessment standard has limited its development and utilization as well [5], which indicated that a comprehensive chemical profiling of *Y. schidigera* was urgently needed.

In order to solve this problem, preliminary studies of the chemical composition of this plant have been conducted in our lab [1,6], and a number of standards for our follow-up research have been prepared. Besides, although liquid chromatography-mass spectrometry (LC-MS) seems to be a good choice for chemical composition characterization, the complexity of plant ingredients always leads to some ignorance about low-level substances, which would result in a chemical composition analysis far from comprehensive [7,8]. Thus, according to the literature, a strategy using multi-phase liquid chromatography (MPLC) combined with MS was considered to improve the chromatographic peak capacity [9].

In this paper, an extract of *Y. schidigera* was separated by a reversed phase chromatography T_3_ column after pretreating with a normal phase chromatography silica column, then analyzed by an ESI-Q-Exactive-Orbitrap MS in positive-ion mode. At the same time, the chromatographic retention behaviors and MS/MS cleavage patterns were summarized. In the light of these summaries, 110 spirostanol saponins were identified in the extract by the MPLC-MS/MS method, and a comprehensive characterization of its spirostanol saponins was set up. Then the verification of the characterizations was accomplished by targeted isolation of ten novel spirostanol saponins (including three pairs of them being (25*R/S*)-spirostanol saponin isomer mixtures).

## 2. Results and Discussion

Through comparing the separation effects of different columns (HSS C_18_, T_3_, CSH FP and BEH C_18_), mobile phase compositions (MeOH-H_2_O, ACN-H_2_O, and MeOH-ACN-H_2_O), pH values (0.1% FA and 0.01% FA), and the column temperatures (30 °C and 35 °C) (Appendix A), the optimal analytical condition for UHPLC was determined as shown in Section 3.2.3. As the composition of YS is complex, in order to solve the severely overlapping chromatographic peaks, the YS extract was pre-treated with a normal-phase silica gel column before analyzing with the T_3_ column (Figure 1).

### 2.1. Superiority of MPLC-MS than Direct LC-MS in the Identification of Spirostanol Saponins from YS

MPLC-MS, combining the advantages of the large peak capacity of MPLC and high sensitivity of mass spectrometry, had a wide range of applications, especially in the systematic chemical composition characterization of TCMs [7,9]. After YS was separated by silica gel column chromatography, the response of some chromatographic peaks was observed to be noticeably improved and some trace peaks were enriched (Appendix A).

#### 2.1.1. Separation and Selectivity of MCI and Silica Gel Column

Through preliminary analysis, the saponins of *Y. schidigera* were mainly seen in the silica gel fractions 6–9 of YS (YSESs6–9) and MCI gel fractions 7–9 of YS (YSEMs7–9). The Sieve statistical results indicated that YSEMs7–9 possessed less peaks than YSESs6–9 (YSEMs7–9: 1024 peaks detected; YSESs6–9: 1301 peaks detected). Besides, for example, the ion peak of *m*/*z* 741.4420 could be found in the BPC of YSES6, but not in YSEMs7–9 (Appendix A). This result suggested that MPLC-MS technique using silica gel column chromatography as a preliminary separation method could result in a better orthogonal effect than MCI.

#### 2.1.2. Separation and Selectivity of MCI and Silica Gel Column

The Sieve statistical results indicated that MPLC-MS using silica gel chromatography as preliminary separation method resulted in many more peaks than direct LC-MS (YS: 486 peaks detected; YSESs6–9: 1301 peaks detected). For example, by comparing the extracted ion chromatograms of *m*/*z* 755.4223 in YS and YSES6, it could be indicated that (25*R*)-Yucca spirostanoside E_4_ (37.76 min, *m*/*z* 755.4223) was only found in the silica gel column separation extract (Appendix A).

### 2.2. Structural Elucidation of Spirostanol Saponins from YSESs by MPLC-MS

#### 2.2.1. Structural Identification of Known Spirostanol Saponins by Comparing with Standard References

Through comparing with standard references (Appendix A), peaks **5**, **7**, **16**, **17**, **26**, **30**–**33**, **35**, **37**, **39**, **44**, **46**, **47**, **52**, **56**, **57**, **60**, **61**, **66**, **72**, **75**, **83**–**86**, **97**, **100**, **104**, **105** were unambiguously identified (Table 1).

#### 2.2.2. Rationale for Structural Characterization

According to the references [1,6,10,12,18], the spirostanol saponins from YS were composed by ten kinds of spirostanol aglycons (YS-1–YS-10, Figure 2) linked with nine kinds of glycosyl groups (***A*_1_**–***A*_9_**, Figure 2, Appendix A) which was made up by glucopyranosyl (Glc), galactopyranosyl (Gal) and xylopyranosyl (Xyl) at C-3 of the aglycons. Among them, YS-7, YS-9 and YS-10 were only mentioned in references and haven′t been obtained by our lab. The occurrence probability of these three kinds of glycosyl groups was Glc > Gal/Xyl.

At the same time, the glycosyls directly linked with aglycons (DG) were limited to Glc and Gal, and whose 2′, 3′ and/or 6′-postions were easily substituted by other kinds of glycosyls, and the characteristic types are listed in Table 1.

When YS-1–YS-3 and YS-6–YS-8 were glycosylated, both Glc and Gal could be directly linked to the C-3 of aglycon, and Glc substitute was more likely to occur, while YS-4, YS-5, YS-9, YS-10 (hydroxyl substituted at C-2) were prone to be glycosylated by Gal directly at C-3 [5,18].

##### Study on MS/MS Cleavage Pattern of Spirostanol Saponins from YSESs

According to what we have summarized, the aglycons of the spirostanol saponins from YSESs were usually substituted by -OH, thus dehydration reactions often happened, and a series of neutral 18 Da losses were produced. Moreover, the E-ring cleavage of ∆^25(27)^ spirostanol saponins was prone to lose 112 Da, forming a [M-C_6_H_8_O_2_ + H]^+^ fragment ion, or lose 142 Da to form a [M-C_8_H_14_O_2_ + H]^+^ fragment ion, while the 25-CH_3_ spirostanol saponins could lose 114 Da to yield a [M-C_6_H_10_O_2_ + H]^+^ fragment ion, or 144 Da was lost, forming a [M-C_8_H_16_O_2_ + H]^+^ fragment ion. The different *m*/*z* as well as the different kinds of cleavage fragment ions (Figure 3, Appendix A) could be used to quickly identify the different types of spirostanol saponins from YSESs. The two pairs of aglycons YS-2 and YS-4 which exhibited the same molecular and fragment ion composition, could be distinguished by observing the relative intensity of the *m*/*z* 283.2420 ion. In detail, because of the 12-OH substitution of YS-2, in its MS/MS spectrum, the intensity of *m*/*z* 283.2420 ion was stronger than that of *m*/*z* 289.2162, while it was opposite of what was observed in the spectrum of 12-H_2_ substituted YS-4. This phenomenon has been observed in the spectrum of YS-9 as well. Although a sample possessing the YS-7 aglycon was lacking, we designed a targeted separation, in order to verify the above rule.

##### Study on MS/MS Cleavage Pattern of Spirostanol Saponins from YSESs

By comparing the chromatographic elution order of reference standards (peaks **16**, **17**, **31**, **32**, **35**, **40**, **41**, **51**, **52**, **60**, **66**, **83**–**86**, **100**, **104**), it was found that the retention times of this class of compounds were mainly affected by the substitution type of the C-12 and C-25 of the aglycon, the stereochemistry of 25-CH_3_, as well as the type of substituted glycosyl (Appendix A).

#### 2.2.3. Structural Elucidation of Spirostanol Saponins from YSESs

Using the rules summarized above, another seventy-nine spirostanol saponins have been tentatively identified. In this section, the tentative identification of ten novel 25(*R*/*S*)-spirostanol saponins would be described in detail. What′s more, the speculation would be verified by their subsequent targeted isolation.

In the MS/MS spectra of peak **10** (*m*/*z* 903.4550), characteristic ions at *m*/*z* 447.3109, 411.2903, 393.2797, 333.2435, 315.2324, 285.1842 could be observed (Table 1, Appendix A). The relative molecular mass of its aglycon was 2 Da more than that of YS-5. The ions of *m*/*z* 333.2435, 315.2324, 285.1842 formed by the cleavage of E ring were similar to those of YS-5 as well, which indicated that the A–E ring was the same as in YS-5. As a result, the aglycon of peak **10** was tentatively assumed to be 25(*R*/*S*)-5β-spirostan-2β,3β-ol-12-one spirostanol. It was speculated that its glycosyl substituent consisted of two molecules of hexose and one pentose molecule because of its neutral loss of 294 Da and 162 Da. According to the biosynthetic pathway of *Y. schidigera*, the structure of peak **10** could be tentatively speculated to be either 25(*R*/*S*)-5β-spirostan-2β,3β-ol-12-one-3-*O*-xylopyranosyl(1→3)-[glucopyranosyl(1→2)]-glucopyranoside or 25(*R*/*S*)-5β-spirostan-2β,3β-ol-12-one-3-*O*-xylo- pyranosyl(1→3)-[glucopyranosyl(1→2)]-galactopyranoside. In order to clarify the correctness of above speculation, a targeted separation was conducted, and peak **10** was finally unambiguously identified as a novel compound, named as 25(*R*)-5β-spirostan-2β,3β-ol-12-one-3-*O*-xylopyranosyl(1→3)-[glucopyranosyl(1→2)]-galactopyranoside.

In the MS/MS spectrum of peaks **14** (*m*/*z* 889.4772), **15** (*m*/*z* 889.4772), **34** (*m*/*z* 727.4261), **36** (*m*/*z* 727.4261), **40** (*m*/*z* 595.3868), and **41** (*m*/*z* 595.3866), characteristic ions at *m*/*z* 433.3312, 397.3101, 379.2995, 301.2526, 289.2162, 283.2420, 271.2056, 253.1951 could be observed (Table 1, Appendix A). According to what we have summarized in the Section “*Study on MS/MS Cleavage Pattern of Spirostanol Saponins from YSESs*”, as the intensities of all their *m*/*z* 283.2420 peaks were stronger than those of the *m*/*z* 289.2162 ones, their aglycon was proposed to be the 12-OH isomer of YS-9, and in particular, it might be YS-7. According to their neutral ion losses, peak **14** [*m*/*z* 889.4772, 595.3867 (294 Da loss) and 433.3323(162 Da loss)], peak **15** [*m*/*z* 889.4767, 595.3850 (294 Da loss) and 433.3330 (162 Da loss)], peak **34** [*m*/*z* 727.4261, 595.3719 (132 Da loss) and 433.3330 (162 Da loss)], peak **36** [*m*/*z* 727.4261 and 433.3314 (294 Da loss)], peak **40** [*m*/*z* 595.3864 and 433.2562 (162 Da loss)], peak **41** [*m*/*z* 595.3864 and 433.2563 (162 Da loss)] were indicated to be substituted by two hexoses and one pentose, two hexoses and one pentose, one hexose and one pentose, one hexoseand one pentose, one hexose and one hexose, respectively. The MS/MS spectra of peaks **14** and **15**, **34** and **36**, **40** and **41** were almost identical, and their retention times were also very close, therefore, they were conjectured to be three pairs of (25*R*/*S*)-isomers with the YS-7 aglycon. Considering the biosynthetic pathway of *Y. schidigera* as well as the chromatographic elution order, peaks **14**, **15**, **34**, **36**, **40** and **41** were speculated to be 25(*S*)-5β-spirostan-3β,12β-diol 3-*O*-xylopyranosyl(1→3)-[glucopyranosyl(1→2)]-glucopyranoside (**14**), 25(*R*)-5β-spirostan-3β,12β-diol 3-*O*-xylopyranosyl(1→3)-[glucopyranosyl(1→2)]-glucopyranoside (**15**), 25(*S*)-5β-spirostan-3β,12β-diol 3-*O*-xylopyranosyl(1→3)-glucopyranoside (**34**), 25(*R*)-5β-spirostan-3β,12β-diol 3-*O*-xylopyranosyl(1→3)-glucopyranoside (**36**), 25(*S*)-5β-spirostan-3β,12β-diol 3-*O*-glucopyranoside (**40**) and 25(*R*)-5β-spirostan-3β,12β-diol 3-*O*-glucopyranoside (**41**), respectively. Furthermore, they were all novel compounds, and targeted for separation to verify the above speculation.

For peaks **27** (*m*/*z* 887.4648), **38** (*m*/*z* 887.4665), **51** (*m*/*z* 755.4223), characteristic ions at *m*/*z* 431.3156, 413.3050, 395.2945, 377.2839, 299.2369, 281.2264, 269.1900 could be observed in their MS/MS spectra (Table 1, Appendix A), which indicated that their aglycon was YS-8. The neutral losses of peak **27** [*m*/*z* 887.4518, 755.4220 (132 Da loss), 593.3700 (162 Da loss) and 431.3167 (162 Da loss)], peak **38** [*m*/*z* 887.4665, 593.3695 (294 Da loss) and 431.3172 (162 Da loss)], and peak **51** [*m*/*z* 755.4230, 593.3699 (162 Da loss) and 431.3173 (162 Da loss)] suggested they were substituted by two hexose molecules and one pentose molecule, two hexose molecules and one pentose molecule and two hexose molecules, respectively. By referring to references as well as comparing them to reference standards, peaks **24** and **32** were the isomers of peaks **27** and **38**, which have been identified to be 25(*R*)-5β-spirostan-3β-ol-12-one 3-*O*-xylopyranosyl(1→3)-[glucopyranosyl(1→2)]-galactopyranoside (**24**) and 25(*R*)-5β-spirostan-3β-ol-12-one 3-*O*-xylopyranosyl(1→3)-[glucopyranosyl(1→2)]-glucopyranoside (**32**), respectively. Thus peaks **27** and **38** were tentatively speculated to be YS-8 substituted by two hexose molecules and one pentose molecule. During the targeted separation and structure elucidation, they were unambiguously identified as 25(*R*)-5β-spirostan-3β-ol-12-one 3-*O*-apiofuranosyl(1→3)-[glucopyranosyl(1→2)]-glucopyranoside (**27**) and 25(*R*)-5β-spirostan-3β-ol-12-one 3-*O*-xylopyranosyl(1→3)-[glucopyranosyl(1→6)]-glucopyranoside (**38**), respectively. Similarly, peak **51** was unambiguously identified as 25(*R*)-5β-spirostan-3β-ol-12-one 3-*O*-glucopyranosyl(1→3)-glucopyranoside (**51**). Peaks **27**, **38** and **51** are all novel compounds.

Similar MS/MS cleavage patterns and chromatographic elution orders as well as the comparisons with references (Table 1) were used to identify another seventy-nine spirostanol saponins, among which, thirty of them were potentially new ones.

#### 2.2.4. Targeted Separation of Peaks **10**, **14**, **15**, **27**, **34**, **36**, **38**, **40**, **41**, **51**

For the purpose of verifying the above speculations, targeted separation of peaks **10**, **14**, **15**, **27**, **34**, **36**, **38**, **40**, **41**, **51** was carried out. In the light of the MPLC-MS analysis results, peak **10** was found to be enriched in the “*Preparation of the YSESs Test Solutions*” fraction 9; peaks **14**, **15** and **38** were found to be enriched in the “*Preparation of the YSESs Test Solutions*” fraction 8; peak **27** was found to be enriched in the “*Preparation of the YSESs Test Solutions*” fraction 7; peaks **34**, **36**, **40**, **41** and **51** were found to be enriched in the “*Preparation of the YSESs Test Solutions*” fraction 6. Then silica gel, ODS CC and preparative HPLC (pHPLC) were used to isolate these fractions and the spirostanol saponins **10**, **14**/**15**, **27**, **34**/**36**, **38**, **40**/**41**, **51** were thus obtained.

##### (25*R*)-5β-Spirostan-2β,3β-diol-12-one 3-*O*-β-d-xylopyranosyl(1→3)- [β-d-glucopyranosyl(1→2)]-β-d-galactopyranoside (**10**)

A white powder with negative optical rotation ([α]_D_^25^ –7.7, MeOH). Its molecular formula, C_44_H_70_O_19_, was deduced by positive-ion ESI-Q-Exactive-Orbitrap MS (*m*/*z* 903.4550 [M + H]^+^; calcd for C_44_H_71_O_19_, 903.4584). Acid hydrolysis yielded d-galactose, d-glucose, and d-xylose [1]. The ^1^H- and ^13^C-NMR (Appendix A) spectra suggested the presence of one β-d-galactopyranosyl [δ 4.98 (1H, d, *J* = 7.5 Hz, H-1′)], one β-d-glucopyranosyl [δ 5.57 (1H, d, *J* = 7.5 Hz, H-1”)], together with one β-d-xylopyranosyl [δ 5.23 (1H, d, *J* = 8.0 Hz, H-1”)]. The ^13^C-NMR signals attributed to the sugar moieties of **10** were consistent with those of yucca spirostanoside D_1_ (5β-spirost-25(27)-en-2β,3β-diol-12-one 3-*O*-β-d-xylopyranosyl(1→3)-[β-d-glucopyranosyl(1→2)]-β-d-galactopyranoside) [1]. A combination of HSQC, HSQC-TOCSY, and ^1^H ^1^H COSY spectra analysis led to the assignment of the three glycosyl units. Forty-four carbon signals have been observed in its ^13^C-NMR spectrum. In addition to the carbon signals represented by the above three glycosyl units, the other 27 signals indicated **10** was a spirostane-type steroid saponin. Its ^1^H- and ^13^C-NMR spectra showed signals for two tertiary methyl groups at δ_H_ 0.98, 1.08 (3H each, both s, H_3_-19, 18), one secondary methyl group at δ_H_ 1.35 (3H, d, *J* = 7.0 Hz, H_3_-21), one oxygenated methylene group at δ_H_ [3.50 (1H, dd, *J* = 10.5, 10.5 Hz), 3.59 (1H, m, overlapped), H_2_-26], three oxygenated methine protons at δ_H_ [3.75, 4.40 (1H each, both m, overlapped, H-2, 3), 4.53 (1H, q like, ca. *J* = 9 Hz, H-16)], together with one carboxyl carbon at δ_C_ 212.7 (C-12). The ^1^H ^1^H COSY spectrum of **10** suggested the presence of the three partial structures indicated by bold lines in Figure 4. The planar structure of the aglycon was determined based on the key HMBC correlations from H_3_-18 to C-12–14, C-17; H_3_-19 to C-1, C-5, C-9, C-10; H_3_-21 to C-17, C-20, C-22; H_2_-26 to C-22; H_2_-27 to C-24–26. Moreover, the connection positions of glycosyl units were determined by the long-range correlations from H-1′ to C-3, H-1” to C-2′, H-1′′′ to C-3′ observed in the HMBC experiment. The ^1^H- and ^13^C-NMR data for the protons and carbons in the A–E rings were identical to those of Yucca spirostanoside D_1_ [1], thus the configuration of the A–E rings was determined. The comparison results of its ^13^C-NMR data for F ring (C-22–26) and C-27 [δ 17.3 (C-27), 29.2 (C-24), 30.5 (C-25), 31.8 (C-23), 67.0 (C-26), 109.3 (C-22)] with those of (25*R*)-5-spirostan [δ17.3 (C-27), 29.2 (C-24), 30.5 (C-25), 31.8 (C-23), 67.0 (C-26), 109.3 (C-22)] [11], further clarified the absolute configuration of C-25. As a result, the structure of **10** was identified as (25*R*)-5β-spirostan-2β,3β-diol-12-one 3-*O*-β-d-xylopyranosyl(1→3)-[β-d-glucopyranosyl(1→2)]-β-d-galactopyranoside.

##### 25(*S/R*)-5β-spirostan-3β,12β-diol 3-*O-*β-d-xylopyranosyl(1→3)-[β-d-glucopyranosyl(1→2)] -β-d-glucopyranoside (**14/15**), 25(*S/R*)-5β-spirostan-3β,12β-diol 3-*O*-β-d-xylopyranosyl(1→3)-β-d-gluco-pyranoside (**34/36**) and 25(*S/R*)-5β-spirostan-3β,12β-diol 3-*O-*β-d-glucopyranoside (**40/41**)

The ^1^H- and ^13^C-NMR (Appendix A, respectively) spectra of **14/15**, **34/36** and **40/41** indicated that the three pairs of spirostanol saponins possessed the same aglycon, which was very similar to that of **10**, 25(*R*/*S*)-5β-spirostan-3β,12β-diol. The difference was that the 2-OH was lacking in the structures of **14/15**, **34/36** and **40/41**, meanwhile the 12-C=O of **10** has been changed into a 12-OH group. The long-range correlations from H-14, 17 and H_3_-18 to C-12 observed in their HMBC spectra as well as the correlations between H_2_-2 and H-3 displayed in their ^1^H-^1^H COSY spectra verified the correctness of the above speculation. By comparing the C-22–26 and 27 carbon signals of **10**, **14/15**, **34/36** and **40/41** they were determined to be 25*R* and 25*S* isomer mixtures. As their ^1^H- and ^13^C-NMR data were consistent with those of 25(*R* and *S*) schidigera-saponin F_1_ [6], their aglycon was identified as 25(*R*/*S*)-5β-spirostan-3β,12β-diol. Acid hydrolysis yielded d-galactose, d-glucose, and d-xylose; d-glucose and d-xylose; and d-glucose, respectively [1]. According to their ^1^H- and ^13^C- NMR spectra (Appendix A), **14/15** possessed one β-d-galactopyranosyl [δ 4.90 (1H, d, *J* = 8.0 Hz, H-1′)], one β-d-glucopyranosyl [δ 5.65 (1H, d, *J* = 8.0 Hz, H-1”)], together with one β-d-xylopyranosyl [δ 5.29 (1H, d, *J* = 7.5 Hz, H-1′′′)]; **34/36** possessed one β-d-glucopyranosyl [δ 4.90 (1H, d, *J* = 8.0 Hz, H-1′)], and one β-d-xylopyranosyl [δ 5.27 (1H, d, *J* = 7.5 Hz, H-1”)]; **40/41** possessed one β-d-glucopyranosyl [δ 4.92 (1H, d, *J* = 7.5 Hz, H-1′)]. Finally, combining the long-range correlations from H-1′ to C-3, H-1” to C-2′, and H-1′′′ to C-3” for **14/15**; H-1′ to C-3, and H-1” to C-3” for **34/36**; H-1′ to C-3 for **40/41**, their structures were elucidated to be 25(*S*/*R*)-5β-spirostan-3β,12β-diol 3-*O*-β-d-xylopyranosyl(1→3)-[β-d-glucopyranosyl(1→2)]-β-d-glucopyranoside (**14/15**), 25(*S*/*R*)-5β-spirostan-3β,12β-diol 3-*O*-β-d-xylopyranosyl(1→3)-β-d-glucopyranoside (**34/36**) and 25(*S*/*R*)-5β-spirostan-3β,12β-diol 3-*O*-β-d-glucopyranoside (**40/41**), respectively. According to the chromatographic elution order summarized in Appendix A, the retention time of 25(*S*)-spirostanol saponin was shorter than that of 25(*R*)-spirostanol saponin, thus the six peaks were finally identified as 25(*S*)-5β-spirostan-3β,12β-diol 3-*O*-β-d-xylopyranosyl(1→3)-[β-d-glucopyranosyl(1→2)]-β-d-glucopyranoside (**14**), 25(*R*)-5β-spirostan-3β,12β-diol 3-*O*-β-d-xylopyranosyl(1→3)-[β-d-glucopyranosyl(1→2)]-β-d-glucopyranoside (**15**), 25(*S*)-5β-spirostan-3β,12β-diol 3-*O*-β-d-xylopyranosyl (1→3)-β-d-glucopyranoside (**34**), 25(*R*)-5β-spirostan-3β,12β-diol 3-*O*-β-d-xylopyranosyl(1→3)-β-d-glucopyranoside (**36**), 25(*S*)-5β-spirostan-3β,12β-diol 3-*O*-*β*-d-glucopyranoside (**40**) and 25(*R*)-5β-spirostan-3β,12β-diol 3-*O*-β-d-glucopyranoside (**41**), respectively.

##### (25*R*)-5β-spirostan-3β-ol-12-one 3-*O-*β-d-apiofuranosyl(1→3)-[β-d-glucopyranosyl(1→2)]-β-d-glucopyranoside (**27**), *(25R*)-5β-spirostan-3β-ol-12-one 3-*O-*β-d-xylopyranosyl(1→3)-[β-d-glucopyranosyl-(1→6)]-β-d-glucopyranoside (**38**) and (25*R*)-5β-spirostan-3β-ol-12-one 3-*O-*β-d-glucopyranosyl(1→3)-β-d-glucopyranoside (**51**)

Compounds **27**, **38** and **51** were all white powders with negative optical rotations. They were determined to possess the same aglycon by comparing their ^1^H- and ^13^C-NMR (Appendix A, C_5_D_5_N) data with each other, that was identical to the speculation in Section 2.2.3. The ^1^H-, ^13^C-NMR and 2D-NMR (^1^H ^1^H COSY, HSQC, HMBC) spectra of the aglycon indicated that it was similar to that of **14/15**, the difference being that the 12-OH group was lacking while a 12-C=O appeared in the structures of these three compounds. This speculation was proven by the long-range correlations from H-14, H-17 and H_3_-18 to C-12 observed in their HMBC spectra (Figure 4). In addition, the pentose signals of **27** and **38** were changed as well. According to the literature, the pentose in the structures of **27** and **38** was elucidated to be β-d-apiofuranosyl [19]. The HSQC, and HSQC-TOCSY combined with COSY spectra were used for the assignment of glycosyl units. Their connections were identified by the long-range correlations from H-1′ to C-3, H-1” to C-2′, H-1′′′ to C-3′ observed in their HMBC spectra. For **51**, there were thirty-nine carbon signals in the ^13^C-NMR spectrum. Aside from the twenty-seven carbons belonging to the aglycon, twelve carbons were left to assign. Acid hydrolysis only yielded d-glucose [1]. The terminal hydrogen signals [δ 4.89 (1H, d, *J* = 8.0 Hz, H-1′), 5.35 (1H, d, *J* = 8.0 Hz, H-1”)] suggested the presence of two β-d-glucopyranosyls. In its HMBC spectrum, long-range correlations from H-1′ to C-3, H-1” to C-3′ could be observed. As a result, their structures were finally elucidated to be (25*R*)-5β-spirostan-3β-ol-12-one 3-*O*-β-d-apiofuranosyl-(1→3)-[β-d-glucopyranosyl(1→2)]-β-d-glucopyranoside (**27**), (25*R*)-5β-spirostan-3β-ol-12-one 3-*O*-β-d-xylopyranosyl(1→3)-[β-d-glucopyranosyl(1→6)]-β-d-glucopyranoside (**38**) and (25*R*)-5β- spirostan-3β-ol-12-one 3-*O*-β-d-glucopyranosyl(1→3)-β-d-glucopyranoside (**51**), respectively. The appearance of a β-d-apiofuranosyl moiety provided new possibilities for speculation about the structure of other compounds.

These results not only confirmed the speculation based on chromatographic retention behaviors and the MS/MS cleavage patterns, but also made supplemented the assignment of the spirostanol saponin glycosyl substituents of *Y. schidigera*.

In general, based on the phytochemistry researches reported before, this study accomplished a comprehensive chemical profiling of the spirostanol saponins in *Y. schidigera*, especially, the targeted isolation experiment made the analysis results more convincing. Moreover, the rules summarized in this paper also provided more accurate references to identify this kind of compounds.

## 3. Materials and Methods

### 3.1. Standard References for LC-MS/MS Research

Thirty-one spirostanol saponins, namely Yucca spirostanosides A_1_, A_2_, B_1_, B_2_, B_3_, C_1_, C_2_, C_3_, D_1_, schidigera-saponin A_1_, schidigera-saponin A_3_, 5β-spirost-25(27)-en-3β-ol-12-one 3-*O*-β-d-glucopyranosyl(1→2)-*O*-[β-d-glucopyranosyl(1→3)]-β-d-glucopyranoside, schidigera-saponin C_2_, schidigera-saponin C_1_ [1]; (25*R*)-Yucca spirostanoside E_1_, (25*S*)-Yucca spirostanoside E_1_, (25*R*)-Yucca spirostanoside E_2_, (25*S*)-Yucca spirostanoside E_2_, (25*R*)-Yucca spirostanoside E_3_, (25*S*)-Yucca spirostanoside E_3_, (25*R*)-5β-spirostan-3β-ol 3-*O*-β-d-glucopyranoside, asparagoside A, 25(*R*)-schidigera-saponin D_5_, 25(*S*)-schidigera-saponin D_5_, 25(*R*)-schidigera-saponin D_1_, 25(*S*)-schidigera-saponin D_1_ [6]; 25(*S*)-schidigera-saponin D_3_, 25(*S*)-schidigera-saponin E_1_, YS-VII, 25(*S*)-schidigera-saponin F_2_, 25(*S*)-schidigera-saponin F_1_ isolated from *Y. schidigera* by our lab were used as references. Their purities were >98%.

### 3.2. General Experimental Procedures, Materials and Methods

#### 3.2.1. Materials and Reagents

Stems of *Y. schidigera* were collected from in the state of FL (USA) and identified by Dr. Li Tianxiang (The Hall of TCM Specimens, Tianjin University of TCM, Tianjin, China). A voucher specimen was deposited at the Academy of Traditional Chinese Medicine of Tianjin University of TCM (No. 20160301).

Acetonitrile (ACN), methanol (MeOH), formic acid (FA) of HPLC grade (Thermo, Waltham, MA, USA), ultra-pure water prepared with a Milli-Q purification system (Millipore, Billerica, MA, USA) and different HPLC columns [ACQUITY UPLC^®^ BEH C_18_ (1.7 µm, 2.1 × 100 mm, Waters, Milford, MA, USA), ACQUITY UPLC^®^ T_3_ (1.8 μm, 2.1 × 100 mm, Waters), ACQUITY UPLC^®^ HSS C_18_ (1.8 μm, 2.1 × 100 mm, Waters), ACQUITY UPLC^®^ C_18_ (1.8 μm, 2.1 × 100 mm, Waters)] were used for LC/MS analysis.

Analytical grade ethanol (EtOH), dichloromethane (CH_2_Cl_2_), methanol (MeOH) and macroporous resin D101 (Haiguang Chemical Co., Ltd., Tianjin, China), silica gel (48–75 µm, Haiyang Chemical Reagent Factory, Qingdao, China), ODS (40–63 µm, YMC Co., Ltd., Tokyo, Japan), MCI GEL CHP 20P (75–150 µm, Mitsubishi Chemical Holdings, Tokyo, JPN) were used for the preparation of *Y. schidigera* 70% EtOH extract (YS), silica gel fractionations of YS (YSESs), MCI gel fractionations of YS (YSEMs) test solutions and the separation of new spirostanol saponins. Preparative high-performance liquid chromatography (PHPLC) columns (Cosmosil 5C_18_-MS-II (20 mm i.d. × 250 mm, Nacalai Tesque, Inc., Kyoto, Japan), Wacopak Navi C_30_-5 (7.5 mm i.d. × 250 mm, Wako Pure Chemical Industries, Ltd., Osaka, Japan), and Cosmosil PBr (20 mm i.d. × 250 mm, Nacalai Tesque, Inc.) were used for the separation of new spirostanol saponins.

Optical rotations were measured on a Rudolph Autopol^®^ IV automatic polarimeter (l = 50 mm) (Rudolph Research Analytical, Hackettstown, NJ, USA). IR spectra were recorded on a Varian 640-IR FT-IR spectrophotometer (Varian Australia Pty Ltd., Mulgrave, Australia). Positive-ion mode ESI-Q-Exactive-Orbitrap MS were obtained on a Thermo Q Exactive Orbitrap MS spectrometer (Thermo). NMR spectra were determined on a Bruker 500 MHz NMR spectrometer (Bruker BioSpin AG Fällanden, Switzerland) at 500 MHz for ^1^H- and 125 MHz for ^13^C-NMR (internal standard: TMS).

#### 3.2.2. Sample Preparation

##### Preparation of Standard Solutions

Standard test solutions of above-mentioned references were prepared in MeOH at a final concentration of approximately 100 ng/mL. All stock solutions were stored at 4 °C in darkness and brought to room temperature before use.

##### Preparation of YS Test Solutions

An aliquot of 1 kg dried powder of *Y. schidigera* stems was extracted under reflux in 8, 6, 6 L 70% ethanol (*v*/*v*) for 3, 2, 2 h, respectively. The extract was combined and 50 mL was filtered with 0.22 µm microporous membrane to obtain YS test solutions. The rest of them was reserved as YS stock solution. YS test solutions were all stored at 4 °C in darkness and brought to room temperature before use.

##### Preparation of the YSEMs Test Solutions

The section “*Preparation of YS Test Solutions*” YS stock solution was evaporated under reduced pressure to obtain YS extract (160.0 g), which (150.0 g) was dissolved in 8 L of water and separated by D101 macroporous adsorption resin column (H_2_O→95% EtOH) to obtain H_2_O eluates (80.2 g) and 95% EtOH eluate (YSE, 60.7 g), respectively. YSE (30.0 g) were subjected to normal pressure MCI gel (200.0 g, 4.5 × 18 cm) column [MeOH-H_2_O (10:90→20:80→30:70→40:60→50:50→60:40→70:30→80:20→100:0, *v*/*v*), fraction volume: 300 mL], nine fractions [YSEM1 (0.85 g), YSEM2 (1.32 g), YSEM3 (2.30 g), YSEM4 (5.45 g), YSEM5 (3.28 g), YSEM6 (1.00 g), YSEM7 (2.07 g), YSEM8 (5.12 g), YSEM9 (8.05 g)] were given. YSEM7–YSEM9 were evaporated to dryness under reduced pressure, and then dissolved with MeOH to get three test stock solutions (5 mg/mL) due to their enrichment of spirostanol saponin. YSEM7–YSEM9 stock solutions were stored at 4 °C in darkness and brought to room temperature before use.

##### Preparation of the YSESs Test Solutions

The section “*Preparation of the YSEMs Test Solutions*” YSE stock solution (30.0 g) was subjected to normal pressure silica gel (200.0 g, 4.5 × 30 cm, fraction volume: 500 mL) column [CH_2_Cl_2_→CH_2_Cl_2_-MeOH (100:1→100:3→100:7→5:1→3:1→2:1→0:100, *v*/*v*)], and [YSES1 (0.22 g), YSES2 (1.06 g), YSES3 (1.61 g), YSES4 (0.88 g), YSES5 (0.69 g), YSES6 (6.91 g), YSES7 (5.57 g), YSES8 (5.85 g), YSES9 (6.66 g)] were obtained. YSES6–YSES9 stock solutions (5 mg/mL) were prepared by using the similar method as YSEM7–YSEM9 and stored at 4 °C in darkness and brought to room temperature before use.

#### 3.2.3. Liquid Chromatography Setup

Separation of spirostanol saponins was performed on a Thermo UltiMate 3000 UHPLC instrument equipped with a quaternary pump, an autosampler. After the optimization of stationary phase (BEH C_18_, HSS C_18_, C_18_ and T_3_), mobile phase (MeOH-H_2_O, ACN-H_2_O, MeOH-ACN-H_2_O), pH (0.1% and 0.01% FA) and column temperature (30 °C and 35 °C) (Figs S1–S4). Samples were separated on a Waters ACQUITY UPLC^®^ T_3_ (2.1 × 100 mm, 1.8 μm) using a mobile phase composed of H_2_O (A) and FA-MeOH-ACN (0.1:50:50, *v*/*v*/*v*) (B) in the gradient program: 0–15 min, 12–24% B; 15–30 min, 24–32% B; 30–42 min, 32–40% B; 42–47 min, 40–60% B; 47–65 min, 60–95% B. An equilibration of 3 min was used between successive injections. The flow rate was 0.3 mL/min, and column temperature was 30 °C. An aliquot of 3 µL of each sample was injected for analysis.

#### 3.2.4. ESI-Q-Exactive-Orbitrap MS High Resolution Tandem Mass Spectrometry and Automatic Components Extraction

Spirostanol saponins identification was carried out by using a Thermo ESI-Q-Exactive-Orbitrap MS (in the positive ESI mode (capillary voltage: 3.2 kV). Ultra-high purity nitrogen (N_2_) and high purity nitrogen (N_2_) were used as the collision gas and the sheath/auxiliary gas, respectively. The ESI source parameters were 350 °C for capillary temperature, 300 °C for ion source heater temperature, 40 arbitrary units for sheath gas (N_2_), 10 arbitrary units for auxiliary gas (N_2_), and the collision energy of the quadrupole ranged between 15 and 45 V were used. The mass range of the Orbitrap analyzer scanner was *m*/*z* 150 to 1500. Selected precursors analyzed more than two times were actively excluded from analysis for 60 s. Monitoring time was 0–65 min. Data recording and processing were performed using the Xcalibur 4.0 software (Thermo Fisher Scientific, Inc., Waltham, MA, USA). The accuracy error threshold was fixed at 5 ppm. Automatic components extraction was accomplished by a Sieve v2.2 SP2 (Thermo Fisher Scientific) with the time range from 0 to 65 min and the BP minimum count at 10,000. Mass resolution is 70,000 and 17,500 for MS and MS^2^, respectively.

#### 3.2.5. Separation and Verification of the New Spirostanol Saponins Speculated by MPLC-MS/MS

The section “*Preparation of the YSESs Test Solutions*” fraction YSES6 (6.0 g) was separated by normal pressure ODS CC (40.0 g, 3 × 8 cm, fraction volume: 50 mL) [MeOH-H_2_O (30:70→40:60→50:50→60:40→70:30→80:20→100:0, *v*/*v*)] and 14 fractions [Fr. 6-1 (150.0 mg), Fr. 6-2 (200.5 mg), Fr. 6-3 (325.2 mg), Fr. 6-4 (326.6 mg), Fr. 6-5 (126.6 mg), Fr. 6-6 (821.3 mg), Fr. 6-7 (435.5 mg), Fr. 6-8 (642.3 mg), Fr. 6-9 (164.8 mg), Fr. 6-10 (492.2 mg), Fr. 6-11 (298.3 mg), Fr. 6-12 (300.5 mg), Fr. 6-13 (542.5 mg), Fr. 6-14 (1100.5 mg)] were obtained. Fraction 6-11 (298.3 mg) was purified by silica gel CC (CH_2_Cl_2_-MeOH (100:7, *v*/*v*)) to obtain 25(*S/R*)-5β-spirostan-3β,12β-diol 3-*O*-β-D-xylopyranosyl(1→3)-β-D-glucopyranoside (**34/36**, 11.7 mg). Fraction 6-11-5 (28.7 mg) was subjected to silica gel CC [CH_2_Cl_2_-MeOH (100:7, *v*/*v*)], and 25(*S/R*)-5β-spirostan-3β,12β-diol 3-*O*-β-d-glucopyranoside (**40/41**, 11.9 mg) was given. Fraction 6-12 (400.5 mg) was purified by PHPLC [MeOH-H_2_O (75:25, *v*/*v*) + 1% HAc, Cosmosil 5C_18_-MS-II column (20 mm i.d. × 250 mm), 9 mL/min] to obtain (25*R*)-5β-spirostan-3β-ol-12-one 3-*O*-β-d-glucopyranosyl(1→3)-β-d-glucopyranoside (**51**, 3.3 mg).

The section “*Preparation of the YSESs Test Solutions*” fraction YSES7 (5.0 g) was subjected to PHPLC [MeOH-H_2_O (80:20, *v*/*v*) + 1% HAc, Cosmosil 5C_18_-MS-II column (20 mm i.d. × 250 mm), 9 mL/min], and 12 fractions [Fr. 7-1 (1010.1 mg), Fr. 7-2 (850.4 mg), Fr. 7-3 (253.8 mg), Fr. 7-4 (223.2 mg), Fr. 7-5 (356.3 mg), Fr. 7-6 (36.5 mg), Fr. 7-7 (223.2 mg), Fr. 7-8 (42.3 mg), Fr. 7-9 (76.2 mg), Fr. 7-10 (492.3 mg), Fr. 7-11 (104.0 mg), Fr. 7-12 (800.1 mg)] were obtained. Fraction 7-5 (356.3 mg) was separated by PHPLC [MeOH-H_2_O (70:30, *v*/*v*) + 1% HAc, Cosmosil 5C_18_-MS-II column (20 mm i.d. × 250 mm), 9 mL/min] to afford (25*R*)-5β-spirostan-3β-ol-12-one 3-*O*-β-d-apiofuranosyl(1→3)-[β-d-glucopyranosyl(1→2)]-β-d-gluco pyranoside (**27**, 9.6 mg).

The section “*Preparation of the YSESs Test Solutions*” fraction YSES8 (5.0 g) was separated by PHPLC [MeOH-H_2_O (80:20, *v*/*v*) + 1% HAc, Cosmosil 5C_18_-MS-II column (20 mm i.d. × 250 mm), 9 mL/min] to yield 17 fractions [Fr. 8-1 (2350.7 mg), Fr. 8-2 (136.2 mg), Fr. 8-3 (370.2 mg), Fr. 8-4 (309.7 mg), Fr. 8-5 (92.4 mg), Fr. 8-6 (212.1 mg), Fr. 8-7 (89.7 mg), Fr. 8-8 (48.1 mg), Fr. 8-9 (60.1 mg), Fr. 8-10 (133.1 mg), Fr. 8-11 (43.1 mg), Fr. 8-12 (10.7 mg), Fr. 8-13 (22.1 mg), Fr. 8-14 (16.3 mg), Fr. 8-15 (40.2 mg), Fr. 8-16 (26.1 mg), Fr. 8-17 (700.0 mg)]. Fraction 8-5 (92.4 mg) was subjected to PHPLC [ACN-H_2_O (38:62, *v*/*v*) + 1% HAc, Cosmosil 5C_18_-MS-II column (20 mm i.d. × 250 mm), 9 mL/min] and PHPLC [MeOH-H_2_O (60:40, *v*/*v*) + 1% HAc, Cosmosil PBr column (20 mm i.d. × 250 mm), 9 mL/min] successively to obtain 25(*S/R*)-5β-spirostan-3β,12β-diol 3-*O*-β-d-xylopyranosyl(1→3)-[β-d-glucopyranosyl(1→2)]-β-d-glucopyranoside (**14/15**, 2.9 mg). Fraction 8-13 (22.1 mg) was isolated by PHPLC [MeOH-H_2_O (78:22, *v*/*v*) + 1% HAc, Wakopak Navi C_30_-5 column (7.5 mm i.d. × 250 mm), 3 mL/min], and (25*R*)-5β-spirostan-3β-ol-12-one 3-*O*-β-d-xylopyranosyl(1→3)-[β-d-glucopyranosyl(1→6)]-β-d-glucopyranoside (**38**, 13.5 mg) was obtained.

The section “*Preparation of the YSESs Test Solutions*” fraction YSES9 (6.0 g) was isolated by PHPLC [MeOH-H_2_O (80:20, *v*/*v*) + 1% HAc, Cosmosil 5C_18_-MS-II column (20 mm i.d. × 250 mm), 9 mL/min] to obtain 16 fractions [Fr. 9-1 (2070.3 mg), Fr. 9-2 (163.2 mg), Fr. 9-3 (115.1 mg), Fr. 9-4 (175.9 mg), Fr. 9-5 (160.1 mg), Fr. 9-6 (175.4 mg), Fr. 9-7 (110.1 mg), Fr. 9-8 (297.2 mg), Fr. 9-9 (112.6 mg), Fr. 9-10 (150.0 mg), Fr. 9-11 (81.7 mg), Fr. 9-12 (147.3 mg), Fr. 9-13 (107.0 mg), Fr. 9-14 (33.6 mg), Fr. 9-15 (75.9 mg), Fr. 9-16 (1650.3 mg)]. Fraction 9-7 (110.1 mg) was separated successively by PHPLC [MeOH-H_2_O (85:15, *v*/*v*) + 1% HAc, Cosmosil 5C_18_-MS-II column (20 mm i.d. × 250 mm), 9 mL/min] and PHPLC [ACN-H_2_O (32:68, *v*/*v*) + 1% HAc, Wakopak Navi C_30_-5 column (7.5 mm i.d. × 250 mm), 3 mL/min] to obtain 25(*R*)-5β-spirostan-2β,3β-ol-12-one-3-*O*-xylopyranosyl(1→3)-[glucopyranosyl (1→2)]-galacto pyranoside (**10**, 9.2 mg).

(25R)-5β-Spirostan-2β,3β-diol-12-one 3-*O*-β-d-xylopyranosyl(1→3)-[β-d-glucopyranosyl(1→2)]-β-d-galactopyranoside (10): White powder; [α]_D_^25^: −7.7° (conc. 0.62, MeOH); IR ν_max_ (KBr) cm^−1^: 3353, 2928, 2872, 1704, 1455, 1377, 1159, 1074, 1043, 983; ^1^H-NMR (C_5_D_5_N, 500 MHz) and ^13^C-NMR (C_5_D_5_N, 125 MHz) data δ: see Appendix A ESI-Q-Exactive-Orbitrap MS positive-ion mode *m*/*z* 903.4550 [M + H]^+^ (calcd for C_44_H_71_O_19_, 903.4584).

25(S/R)-5β-Spirostan-3β,12β-diol 3-*O*-β-d-xylopyranosyl(1→3)-[β-d-glucopyranosyl(1→2)]-β-d-glucopyranoside (14/15): White powder; [α]_D_^25^: −8.0° (conc. 0.34, MeOH); IR ν_max_ (KBr) cm^−1^: 3398, 2930, 2876, 1644, 1453, 1373, 1160, 1077, 1043, 990; ^1^H-NMR (C_5_D_5_N, 500 MHz) and ^13^C-NMR (C_5_D_5_N, 125 MHz) data δ: see Appendix A; ESI-Q-Exactive-Orbitrap MS positive-ion mode *m*/*z* 889.4799 [M + H]^+^ (calcd for C_44_H_73_O_18_, 889.4791).

(25R)-5β-Spirostan-3β-ol-12-one 3-*O*-β-d-apiofuranosyl(1→3)-[β-d-glucopyranosyl(1→2)]-β-d-glucopyranoside (27): White powder; [α]_D_^25^: −10.3° (conc. 0.79, MeOH); IR ν_max_ (KBr) cm^−1^: 3412, 2929, 2874, 1705, 1453, 1161, 1079, 1028, 982; ^1^H-NMR (C_5_D_5_N, 500 MHz) and ^13^C-NMR (C_5_D_5_N, 125 MHz) data δ: see Appendix A; ESI-Q-Exactive-Orbitrap MS positive-ion mode *m*/*z* 887.4658 [M + H]^+^ (calcd for C_44_H_71_O_18_, 887.4635).

25(S/R)-5β-Spirostan-3β,12β-diol 3-*O*-β-d-xylopyranosyl(1→3)-β-d-glucopyranoside (34/36): White powder; [α]_D_^25^: −36.0° (conc. 0.050, MeOH); IR ν_max_ (KBr) cm^−1^: 3396, 2928, 2869, 1646, 1546, 1453, 1372, 1161, 1045, 986; ^1^H-NMR (C_5_D_5_N, 500 MHz) and ^13^C-NMR (C_5_D_5_N, 125 MHz) data δ: see Appendix A; ESI-Q-Exactive-Orbitrap MS Positive-ion mode *m*/*z* 727.4282 [M + H]^+^ (calcd for C_38_H_63_O_13_, 727.4263).

(25R)-5β-Spirostan-3β-ol-12-one 3-*O*-β-d-xylopyranosyl(1→3)-[β-d-glucopyranosyl(1→6)]-β-d-glucopyranoside (38): White powder; [α]_D_^25^: −16.2° (conc. 0.78, MeOH); IR ν_max_ (KBr) cm^−1^: 3423, 2930, 2874, 1705, 1456, 1377, 1164, 1073, 1039, 985; ^1^H-NMR (C_5_D_5_N, 500 MHz) and ^13^C-NMR (C_5_D_5_N, 125 MHz) data δ: see Appendix A; ESI-Q-Exactive-Orbitrap MS Positive-ion mode *m*/*z* 887.4632 [M + H]^+^ (calcd for C_44_H_71_O_18_, 887.4635).

25(S/R)-5β-Spirostan-3β,12β-diol 3-*O*-β-d-glucopyranoside (40/41): White powder; [α]_D_^25^: −44.7° (conc. 0.085, MeOH); IR ν_max_ (KBr) cm^−1^: 3414, 2930, 2871, 1646, 1563, 1452, 1376, 1166, 1061, 1022, 990; ^1^H-NMR (C_5_D_5_N, 500 MHz) and ^13^C-NMR (C_5_D_5_N, 125 MHz) data δ: see Appendix A; ESI-Q-Exactive-Orbitrap MS Positive-ion mode *m*/*z* 595.3847 [M + H]^+^ (calcd for C_33_H_55_O_9_, 595.3841).

(25R)-5β-Spirostan-3β-ol-12-one 3-*O*-β-d-glucopyranosyl(1→3)-β-d-glucopyranoside (51): White powder; [α]_D_^25^: −0.3° (conc. 0.68, MeOH); IR ν_max_ (KBr) cm^−1^: 3388, 2928, 2871, 1706, 1453, 1375, 1159, 1077, 1036, 986; ^1^H-NMR (C_5_D_5_N, 500 MHz) and ^13^C-NMR (C_5_D_5_N, 125 MHz) data δ: see Appendix A; ESI-Q-Exactive-Orbitrap MS Positive-ion mode *m*/*z* 755.4223 [M + H]^+^ (calcd for C_39_H_63_O_14_, 755.4212).

## 4. Conclusions

In conclusion, in order to provide references for the quality assessment standard establishment of *Y. schidigera*, a MPLC-MS/MS analysis technique was used to accomplish the qualitative analysis of the spirostanol saponins from its extract, and a comprehensive characterization method for spirostanol saponins was set up for the first time. This will lay a foundation for the quality evaluation of *Y. schidigera*. On the other hand, the chromatographic retention behaviors and the MS/MS cleavage patterns of spirostanol saponins were summarized. 

In summary, according to the retention time (*t*_R_) and the exact mass-to-charge ratio (*m*/*z*), thirty-one compounds were unambiguously identified by comparing them to references. Meanwhile, the MS/MS fragmentation pattern and chromatographic elution order rules have been generalized by using the standard compounds as references, seventy-nine compounds were tentatively identified and forty of them were potential new ones. Among them, ten were targeted for separation to prove the correctness of our speculations. During the verification process, the appearance of a β-d-apiofuranosyl moiety was found for the first time, which provided new possibilities for speculation about the structure of other compounds. As a result, an accurate and comprehensive chemical composition profiling of the aerial part of the *Y. schidigera* was realized, which lays a foundation for the quality evaluation of the plant.

## Figures and Tables

**Figure 1 molecules-25-03848-f001:**
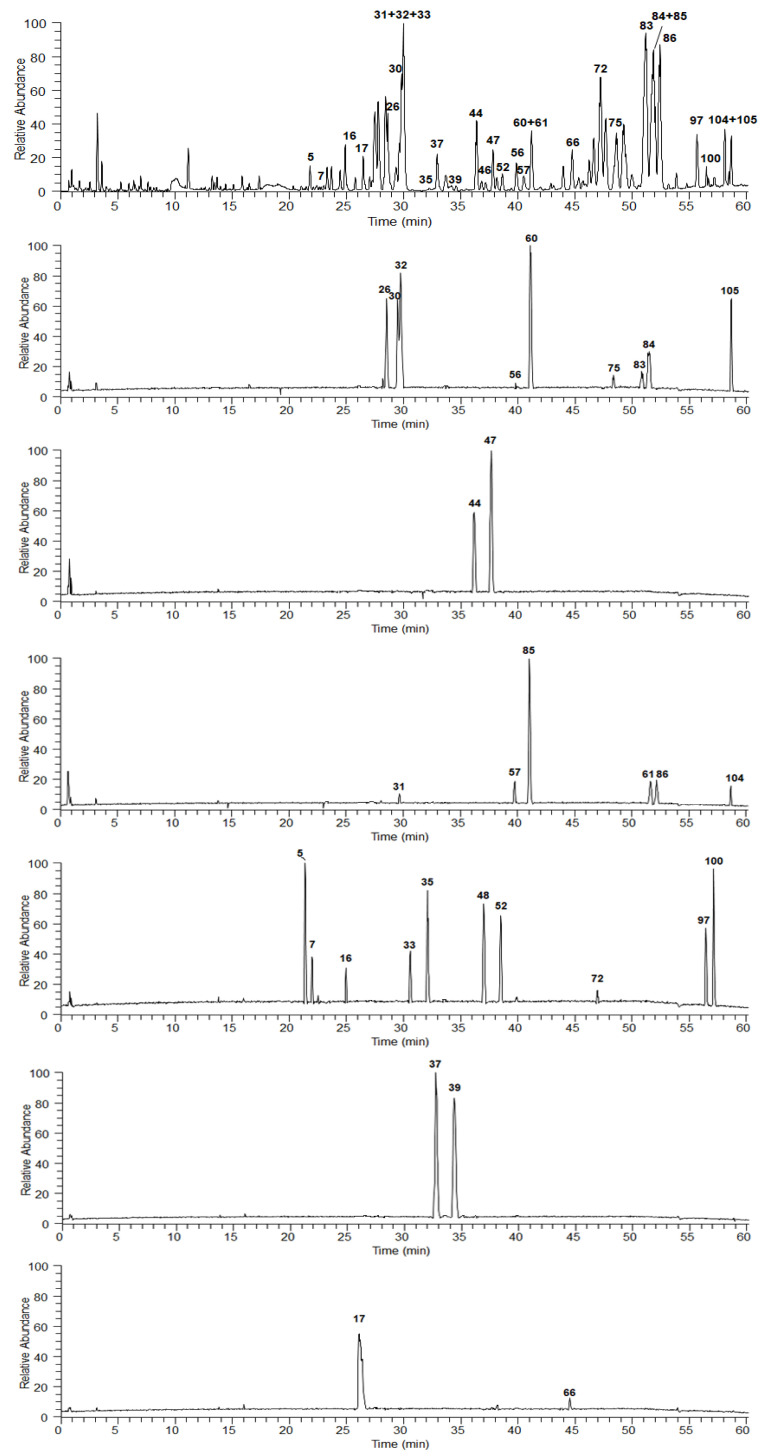
BPC for YS, and standard references **5**, **7**, **16**, **17**, **26**, **30**–**33**, **35**, **37**, **39**, **44**, **46**, **47**, **52**, **56**, **57**, **60**, **61**, **66**, **72**, **75**, **83**–**86**, **97**, **100**, **104**, **105** by UHPLC-ESI-Q-Exactive-Orbitrap MS/MS in positive ion mode.Column: Waters ACQUITY UPLC^®^ T_3_ (2.1 × 100 mm, 1.8 μm). Mobile phase: H_2_O (A) and FA-MeOH-ACN (0.1:50:50, *v*/*v*/*v*) (B). Gradient program: 0–15 min, 12–24% B; 15–30 min, 24–32% B; 30–42 min, 32–40% B; 42–47 min, 40–60% B; 47–65 min, 60–95% B. Flow rate: 0.3 mL/min. Column temperature: 30 °C. Injection volume: 3 µL. ESI-Q-Exactive-Orbitrap MS mode: positive ion mode.

**Figure 2 molecules-25-03848-f002:**
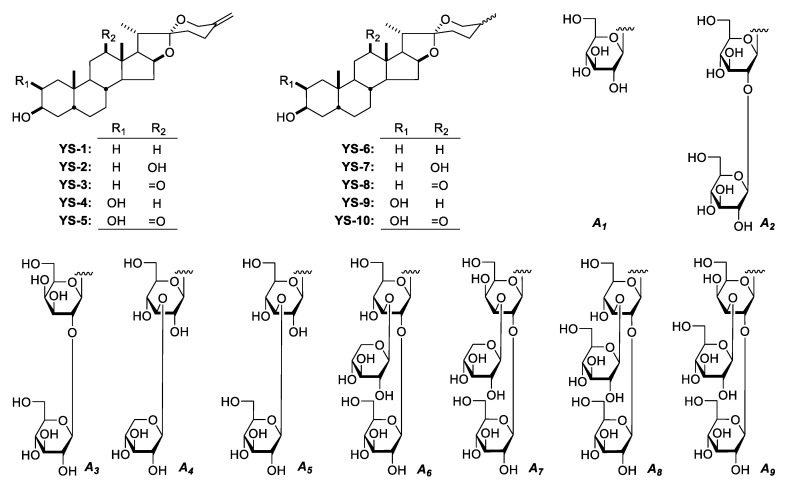
Structures of ten aglycons and nine glycosyls of spirostanol saponins from YS.

**Figure 3 molecules-25-03848-f003:**
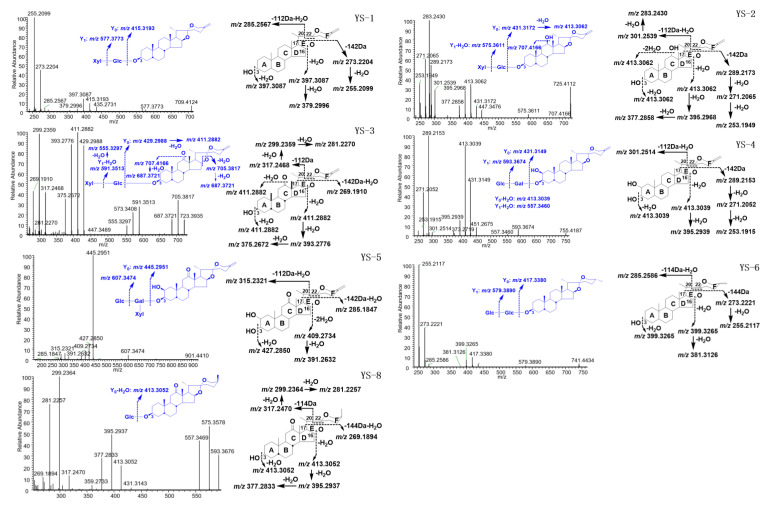
The MS/MS spectrum and proposed fragmentation pathways of YS-1–YS-6 and YS-8.

**Figure 4 molecules-25-03848-f004:**
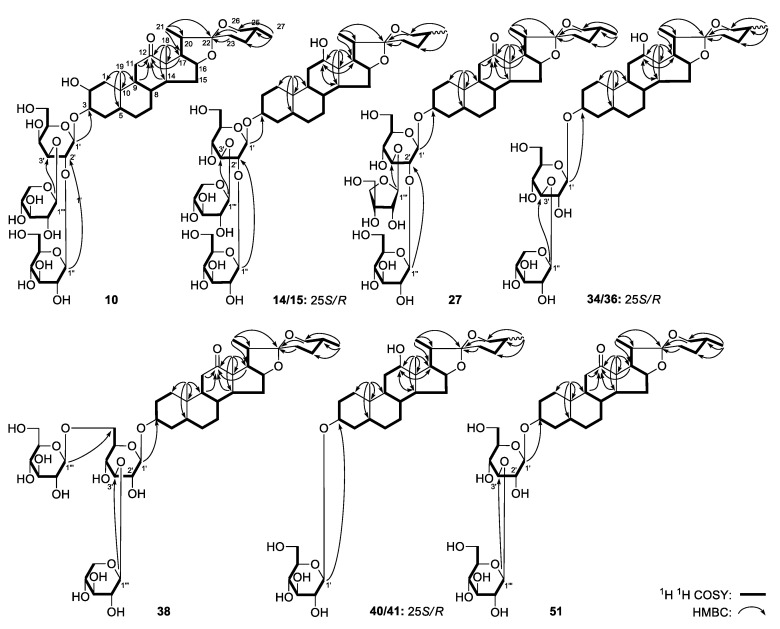
The main ^1^H ^1^H COSY and HMBC correlations of **10**, **14**/**15**, **27**, **34**/**36**, **38**, **40/41**, **51.**

**Table 1 molecules-25-03848-t001:** The qualitative analysis of compounds **1**–**110** by ESI-Q-Exactive-Orbitrap MS

No.	*t*_R_ (min)	Compound	Formula	Adduct Ions	Theoretic	Measure	Diff	MS/MS (*m*/*z*) (Intensity)	Identification Method
**1**	15.85	5β-Spirost-25(27)-en-3β-ol 3-*O*-Glc(1→6)-[Glc(1→2)-Glc(1→3)]-Glc ^#^	C_51_H_82_O_23_	[M + H]^+^	1063.5320	1063.5363	4.04	1063.5363 (1.05), 901.4867 (0.09), 739.4237 (0.31), 577.3756 (2.17), 415.3219 (14.16), 397.3114 (6.38), 379.3023 (1.89), 285.2587 (0.75), 273.2223 (39.86), 255.2115 (100.00)	MS/MS
**2**	16.58	5β-Spirost-25(27)-en-3β-ol 3-*O*-Glc(1→6)-[Glc(1→2)-Xyl(1→3)]-Glc ^#^	C_50_H_80_O_22_	[M + H]^+^	1033.5211	1033.5211	0.00	1033.5211 (1.55), 871.4718 (3.70), 577.3736 (2.29), 415.3219 (25.80), 397.3115 (13.61), 379.3004 (3.75), 285.2583 (1.68), 273.2222 (62.25), 255.2116 (100.00)	MS/MS
**3**	19.76	5β-Spirost-25(27)-en-3β-ol-12-one 3-*O*-Api(1→3)-[Glc(1→2)]-Gal ^#^	C_44_H_68_O_18_	[M + H]^+^	885.4478	885.4502	2.71	885.4502 (6.93), 753.4117 (1.01), 591.3535 (2.01), 429.3017 (100.00), 411.2912 (20.17), 393.2797 (7.65), 299.2379 (3.47), 281.2287 (1.00), 269.1898 (1.29)	MS/MS
**4**	21.16	5β-Spirost-25(27)-en-3β,12β-diol 3-*O*-Xyl(1→3)-[Glc(1→2)]-Gal ^#^	C_44_H_70_O_18_	[M + H]^+^	887.4635	887.4622	−1.46	887.4622 (1.10), 869.4547 (0.73), 755.4313 (0.04), 593.3715 (1.55), 575.3571 (0.21), 431.3175 (15.74), 413.3065 (18.44), 395.2967 (8.47), 377.2862 (2.53), 301.2543 (5.30), 289.2170 (16.59), 283.2433 (100.00), 271.2066 (13.82), 253.1958 (8.88)	MS/MS
**5**	21.19	Yucca spirostanoside D_1_ *	C_44_H_68_O_19_	[M + H]^+^	901.4428	901.4410	−2.00	901.4410 (0.45), 740.1865 (0.34), 607.3474 (4.46), 577.3405 (0.80), 445.2951 (100.00), 427.2850 (15.40), 409.2734 (8.60), 391.2632 (3.36), 373.2473 (0.67), 333.2434 (3.12), 315.2321 (6.24), 302.0273 (0.38), 297.2222 (5.30), 285.1847 (1.26), 279.2111 (1.03), 269.1905 (0.75)	MS/MS, standard reproted by literature [1]
**6**	21.73	5β-Spirost-25(27)-en-3β-ol-12-one 3-*O*-Api(1→3)-[Glc(1→2)]-Glc ^#^	C_44_H_68_O_18_	[M + H]^+^	885.4478	885.4480	0.23	885.4480 (3.83), 753.4049 (2.19), 591.3541 (4.14), 429.3019 (100.00), 411.2903 (21.92), 393.2803 (6.83), 299.2381 (1.57), 269.1895 (0.70)	MS/MS
**7**	22.24	Yucca spirostanoside B_3_ *	C_45_H_72_O_19_	[M + H]^+^	917.4741	917.4745	0.44	917.4745 (1.77), 899.4671 (1.87), 755.4432 (1.38), 593.3698 (7.14), 431.3167 (19.21), 413.3065 (100.00), 395.2961 (10.11), 377.2853 (1.18), 301.2537 (3.22), 289.2166 (15.64), 283.2431 (9.17), 271.20161 (15.46), 253.1956 (6.79)	MS/MS, standard reproted by literature [1]
**8**	22.63	25(*S*)-5β-Spirostan-3β,12β-diol 3-*O*-Glc(1→3)-[Glc(1→2)]-Glc ^#^	C_45_H_74_O_19_	[M + H]^+^	919.4897	919.4919	2.39	900.4546 (0.88), 739.4276 (3.67), 595.3859 (5.33), 577.3771 (4.10), 433.3330 (100.00), 415.3225 (25.37), 397.3117 (7.05), 379.3006 (0.88), 301.2531 (2.67), 289.2171 (7.64), 283.2430 (11.85), 271.2071 (6.41), 253.1956 (4.02)	MS/MS
**9**	22.99	YS XII	C_45_H_74_O_19_	[M + H]^+^	919.4897	919.4919	2.39	900.4546 (0.88), 739.4276 (3.67), 595.3859 (5.33), 577.3771 (4.10), 433.3330 (100.00), 415.3225 (25.37), 397.3117 (7.05), 379.3006 (0.88), 301.2531 (2.67), 289.2171 (7.64), 283.2430 (11.85), 271.2071 (6.41), 253.1956 (4.02)	MS/MS, literature [10]
**10**	23.04	25(*R*)-5β-Spirostan-2β,3β-ol-12-one-3-*O*-Xyl(1→3)-[Glc(1→2)]-Gal	C_44_H_70_O_19_	[M + H]^+^	903.4584	903.4550	−3.76	609.3638 (2.30), 447.3109 (100.00), 429.3012 (23.67), 411.2903 (16.39), 393.2797 (2.64), 333.2435 (4.23), 315.2324 (14.86), 297.2218 (14.03), 285.1842 (1.55), 279.2117 (4.11)	MS/MS
**11**	23.16	5β-Spirost-25(27)-en-3β-ol-12-one 3-*O*-Xyl(1→3)-[Glc(1→2)]-Gal	C_44_H_68_O_18_	[M + H]^+^	885.4478	885.4512	3.84	885.4512 (21.56), 753.4061 (13.27), 591.3555 (11.51), 429.3015 (100.00), 411.2905 (18.53), 393.2809 (8.23), 375.2710 (1.33), 299.2365 (17.25), 281.2261 (11.89)	MS/MS, literature [11]
**12**	23.32	5β-Spirost-25(27)-en-3β,12β-diol 3-*O*-Xyl(1→3)-[Glc(1→2)]-Glc ^#^	C_44_H_70_O_18_	[M + H]^+^	887.4635	887.4638	0.34	887.4638 (2.66), 869.4570 (1.20), 755.4313 (0.59), 593.3729 (1.19), 431.3169 (16.33), 413.3061 (16.54), 395.2949 (7.13), 377.2860 (0.72), 301.2534 (8.14), 289.2169 (17.14), 283.2429 (100.00), 271.2065 (16.39), 253.1966 (7.32)	MS/MS
**13**	23.95	25(*R*/*S*)-5β-Spirostan-3β,12β-diol 3-*O*-Xyl(1→3)-[Glc(1→2)]-Gal ^#^	C_44_H_72_O_18_	[M + H]^+^	889.4791	889.4826	3.93	889.4826 (0.25), 757.4467 (0.67), 595.3856 (0.82), 433.3326 (20.48), 415.3224 (100.00), 397.3108 (8.28), 379.2990 (1.29), 301.2537 (9.22), 289.2169 (14.84), 283.2430 (21.20), 271.2068 (15.87), 253.1960 (8.47)	MS/MS
**14**	24.52	25(*S*)-5β-Spirostan-3β,12β-diol 3-*O*-Xyl(1→3)-[Glc(1→2)]-Glc	C_44_H_72_O_18_	[M + H]^+^	889.4791	889.4772	−2.14	889.4772 (5.85), 757.4364 (1.66), 595.3867 (7.05), 433.3323 (77.23), 415.3209 (93.33), 397.3113 (35.91), 379.2998 (7.35), 301.2537 (37.53), 289.2171 (64.86), 283.2435 (100.00), 271.2072 (71.97), 253.1954 (36.78)	MS/MS
**15**	24.54	25(*R*)-5β-Spirostan-3β,12β-diol 3-*O*-Xyl(1→3)-[Glc(1→2)]-Glc	C_44_H_72_O_18_	[M + H]^+^	889.4791	889.4767	−2.70	889.4767 (7.24), 757.4364 (1.66), 595.3850 (0.81), 433.3330 (87.31), 415.3209 (100.00), 397.3113 (39.52), 379.2998 (16.12), 301.2537 (58.12), 289.2171 (73.16), 283.2435 (98.22), 271.2072 (78.89), 253.1954 (35.94)	MS/MS
**16**	24.76	Yucca spirostanoside C_3_ *	C_45_H_70_O_19_	[M + H]^+^	915.4584	915.4576	−0.87	915.4576 (5.20), 753.4245 (14.19), 591.3542 (3.79), 429.3012 (100.00), 411.2907 (23.96), 393.2802 (7.95), 299.2384 (10.63), 281.2284 (6.60), 269.1950 (1.20)	MS/MS, standard reproted by literature [1]
**17**	26.12	5β-Spirost-25(27)-en-3β-ol-12-one-3-*O*Glc(1→2)-*O*-[Glc(1→3)]-Glc *	C_45_H_70_O_19_	[M + H]^+^	915.4584	915.4620	3.93	915.4620 (3.29), 753.4156 (5.16), 591.3542 (4.97), 429.3015 (100.00), 411.2907 (23.01), 393.2802 (10.50), 299.2384 (14.06), 281.2284 (8.73), 269.1950 (1.28)	MS/MS, standard reproted by literature [1]
**18**	26.26	(25*R*/*S*)-5β-Spirostan-3β-ol-12-one 3-*O*-Glc(1→2)[Glc(1→3)]-Gal ^#^	C_45_H_72_O_19_	[M + H]^+^	917.4741	917.4763	2.40	899.4560 (1.08), 755.4058 (1.02), 593.3698 (7.14), 431.3166 (100.00), 413.3060 (21.21), 395.2961 (10.92), 377.2844 (3.01), 299.2372 (8.79), 281.2269 (6.81), 269.1903 (3.22)	MS/MS
**19**	27.06	25(*R*/*S*)-5β-Spirostan-1β,3β-diol 3-*O*-Xyl(1→3)[Glc(1→2)]-Gal ^#^	C_44_H_72_O_18_	[M + H]^+^	889.4791	889.4766	−2.81	889.4766 (4.20), 757.4364 (1.66), 595.3846 (3.43), 433.3315 (44.14), 415.3234 (62.06), 397.3110 (100.00), 379.3011 (11.82), 301.2541 (13.33), 289.2174 (2.68), 283.2429 (40.52), 271.2062 (48.54), 253.1953 (47.81)	MS/MS
**20**	27.20	Schidigera-saponin B_1_	C_44_H_68_O_18_	[M + H]^+^	885.4478	885.4485	0.79	885.4485 (13.49), 753.3990 (4.73), 591.3539 (8.27), 429.3008 (100.00), 411.2915 (28.92), 393.2801 (13.93), 317.2474 (1.56), 299.2379 (9.89), 287.2002 (1.54), 281.2266 (2.64), 269.1928 (0.24)	MS/MS, literature [5]
**21**	27.27	(25*S*)-3β-[(*O*-Glc(1→2)-Gal)oxy]-5β spirostan-12-one ^#^	C_39_H_62_O_14_	[M + H]^+^	755.4212	755.4200	−1.59	755.4200 (1.45), 737.4117 (2.38), 719.3984 (2.09), 593.3697 (3.68), 575.3598 (0.97), 431.3176 (20.32), 413.3061 (16.68), 395.2963 (100.00), 377.2847 (8.81), 317.2493 (2.66), 299.2377 (12.94), 281.2273 (8.58)	MS/MS
**22**	27.28	(25*R*)-3β-[(*O*-Glc(1→2)-Gal)oxy]-5β spirostan-12-one	C_39_H_62_O_14_	[M + H]^+^	755.4212	755.4213	0.13	755.4213 (2.68), 737.4111 (3.76), 719.3994 (1.47), 593.3711 (10.20), 575.3606 (3.25), 431.3175 (44.57), 413.3069 (100.00), 395.2958 (18.65), 377.2850 (6.00), 317.2488 (6.49), 299.2379 (22.27), 281.2273 (15.26)	MS/MS, literature [12]
**23**	27.47	(25*S*)-5β-Spirostan-3β-ol-12-one 3-*O*-Xyl(1→3)[Glc(1→2)]-Gal ^#^	C_44_H_70_O_18_	[M + H]^+^	887.4635	887.4618	−1.92	887.4618 (5.66), 755.4180 (2.07), 593.3669 (6.91), 431.3166 (100.00), 413.3072 (26.31), 395.2953 (17.04), 377.2848 (3.09), 299.2383 (7.90), 281.2273 (6.10)	MS/MS
**24**	27.88	(25*R*)-5β-Spirostan-3β-ol-12-one 3-*O*-Xyl(1→3)[Glc(1→2)]-Gal	C_44_H_70_O_18_	[M + H]^+^	887.4635	887.4619	−1.80	887.4619 (5.56), 755.4180 (3.35), 593.3671 (6.77), 431.3168 (77.21), 413.3070 (100.00), 395.2954 (18.08), 377.2849 (3.11), 299.2383 (7.95), 281.2275 (6.70)	MS/MS, literature [11]
**25**	28.16	5β-Spirost-25(27)-en-3β-ol-12-one 3-*O*-Glc(1→6)[Glc(1→3)]-Glc ^#^	C_45_H_70_O_19_	[M + H]^+^	915.4584	915.4617	3.60	915.4617 (6.46), 753.4125 (10.03), 591.3551 (9.56), 429.3013 (100.00), 411.2910 (18.14), 393.2773 (2.37), 299.2386 (2.58), 281.2280 (2.68), 269.1909 (1.93)	MS/MS
**26**	28.75	YS-VII *	C_45_H_72_O_19_	[M + H]^+^	917.4741	917.4773	3.49	899.4603 (0.63), 755.4173 (0.55), 593.3710 (9.97), 431.3169 (100.00), 413.3062 (23.14), 395.2959 (14.93), 377.2847 (2.67), 299.2380 (10.61), 281.2270 (7.07), 269.1908 (2.76)	MS/MS, standard isolated by our lab
**27**	28.81	(25*R*)-5β-Spirostan-3β-ol-12-one 3-*O*-β-Api(1→3)-[Glc(1→2)]-Glc	C_44_H_70_O_18_	[M + H]^+^	887.4635	887.4618	−1.92	887.4618 (2.26), 755.4220 (2.13), 593.3700 (10.19), 431.3167 (100.00), 413.3073 (39.61), 395.2947 (23.61), 377.2850 (4.50), 317.2495 (5.20), 299.2387 (19.65), 287.1989 (0.42), 281.2267 (10.08), 269.1905 (4.61), 251.1762 (8.39)	MS/MS
**28**	28.83	25(*R*/*S*)-5β-Spirostan-3β,12β-diol 3-*O*-Xyl(1→3)-Gal ^#^	C_38_H_62_O_13_	[M + H]^+^	727.4263	727.4234	−3.99	727.4234 (19.44), 433.3307 (5.31), 415.3207 (18.74), 397.3099 (3.86), 301.2521 (20.65), 289.2165 (14.43), 283.2428 (100.00), 271.2067 (22.74), 253.1959 (6.20)	MS/MS
**29**	28.85	25(*R*/*S*)-5β-Spirostan-1β,3β-diol 3-*O*-Xyl(1→3)-[Glc(1→2)]-Glc ^#^	C_44_H_72_O_18_	[M + H]^+^	889.4791	889.4826	3.93	889.4826 (4.20), 757.4364 (1.66), 595.3846 (3.43), 433.3322 (37.49), 415.3223 (37.80), 397.3106 (100.00), 379.3000 (12.21), 283.2428 (19.47), 271.2067 (27.54), 253.1958 (36.41)	MS/MS
**30**	29.24	(25*S*)-Yucca spirostanoside E_3_ *	C_39_H_62_O_14_	[M + H]^+^	755.4212	755.4240	3.71	755.4240 (1.74), 737.4130 (6.61), 719.4015 (2.78), 593.3713 (9.00), 575.3615 (2.97), 431.3175 (100.00), 413.3065 (20.10), 395.2957 (14.54), 377.2835 (5.25), 317.2497 (6.17), 299.2381 (20.64), 281.2272 (14.25)	MS/MS, standard reproted by literature [6]
**31**	29.42	(25*R*)-Yucca spirostanoside E_3_ *	C_39_H_62_O_14_	[M + H]^+^	755.4212	755.4202	−1.32	755.4202 (2.79), 737.4119 (2.61), 719.4024 (2.75), 593.3701 (9.37), 575.3602 (2.56), 431.3171 (100.00), 413.3065 (24.54), 395.2972 (16.95), 377.2855 (5.97), 317.2495 (7.55), 299.2380 (27.91), 281.2278 (19.49)	MS/MS, standard reproted by literature [6]
**32**	29.51	25(*S*)-Schidigera-saponin E_1_ *	C_44_H_70_O_18_	[M + H]^+^	887.4635	887.4651	1.80	887.4651 (2.71), 869.4508 (1.62), 755.4108 (1.28), 593.3730 (8.57), 431.3166 (100.00), 413.3054 (30.71), 395.2961 (15.95), 377.2864 (1.86), 299.2373 (14.75), 281.2265 (10.52), 269.1911 (1.89)	MS/MS, standard isolated by our lab
**33**	30.30	Yucca spirostanoside B_2_ *	C_38_H_60_O_13_	[M + H]^+^	725.4107	725.4112	0.69	725.4112 (21.14), 707.4166 (0.28), 593.3740 (2.47), 431.3172 (7.77), 413.3062 (22.17), 395.2968 (18.15), 377.2858 (8.19), 301.2539 (19.11), 289.2173 (36.03), 283.2430 (100.00), 271.2065 (38.65), 253.1960 (24.28)	MS/MS, standard reproted by literature [1]
**34**	31.75	25(*S*)-5β-Spirostan-3β,12β-diol 3-*O*-Xyl(1→3)-Glc	C_38_H_62_O_13_	[M + H]^+^	727.4263	727.4299	4.95	727.4299 (34.55), 595.3719 (2.78), 577.3777 (2.50), 433.3330 (12.89), 415.3212 (20.56), 397.3104 (11.37), 379.3010 (4.10), 301.2542 (26.06), 289.2173 (31.76), 283.2430 (100.00), 271.2069 (28.12), 253.1951 (18.27)	MS/MS
**35**	31.79	Yucca spirostanoside B_1_ *	C_33_H_52_O_9_	[M + H]^+^	593.3684	593.3701	2.86	593.3701 (58.75), 575.3610 (15.96), 431.3178 (3.66), 413.3067 (12.38), 395.2964 (19.97), 377.2858 (9.63), 301.2532 (16.35), 289.2174 (42.46), 283.2431 (100.00), 271.2067 (61.06), 253.1960 (30.48)	MS/MS, standard reproted by literature [1]
**36**	32.01	25(*R*)-5β-Spirostan-3β,12β-diol 3-*O*-Xyl(1→3)-Glc	C_38_H_62_O_13_	[M + H]^+^	727.4263	727.4261	−0.27	727.4261 (39.39), 433.3314 (17.80), 415.3228 (26.74), 397.3110 (24.04), 379.3014 (6.68), 301.2537 (27.67), 289.2178 (34.89), 283.2434 (100.00), 271.2068 (40.31), 253.1959 (8.24)	MS/MS
**37**	32.51	Schidigera-saponin C_1_ *	C_44_H_70_O_18_	[M + H]^+^	887.4635	887.4665	3.38	887.4665 (1.75), 755.4108 (1.41), 593.3685 (3.95), 431.3171 (70.29), 413.3062 (45.24), 395.2954 (9.03), 377.2824 (1.43), 289.2171 (70.00), 283.2418 (1.37), 271.2067 (100.00), 253.1960 (37.96)	MS/MS, standard reproted by literature [1]
**38**	32.51	(25*R*)-5β-Spirostan-3β-ol-12-one 3-*O*-Xyl(1→3)-[Glc(1→6)]-Glc	C_44_H_70_O_18_	[M + H]^+^	887.4635	887.4665	3.38	593.3695 (8.59), 431.3172 (100.00), 413.3065 (45.15), 395.2947 (18.25), 377.2838 (4.77), 299.2377 (8.97), 281.2274 (8.71), 269.1907 (3.33)	MS/MS
**39**	34.06	Schidigera-saponin C_2_ *	C_39_H_62_O_14_	[M + H]^+^	755.4212	755.4232	2.65	755.4232 (3.07), 593.3716 (1.20), 431.3171 (21.93), 413.3065 (33.67), 395.2959 (7.30), 377.2863 (1.18), 289.2172 (60.83), 283.1323 (0.67), 271.2067 (100.00), 253.1959 (24.93)	MS/MS, standard reproted by literature [1]
**40**	33.51	25(*S*)-5β-Spirostan-3β,12β-diol 3-*O*-Glc	C_33_H_54_O_9_	[M + H]^+^	595.3841	595.3864	3.86	595.3864 (68.74), 433.2562 (1.99), 397.3130 (3.88), 379.3032 (4.41), 301.2534 (23.47), 289.2170 (16.30), 283.2432 (100.00), 271.2061 (23.39), 253.1955 (14.22)	MS/MS
**41**	33.69	25(*R*)-5β-Spirostan-3β,12β-diol 3-*O*-Glc	C_33_H_54_O_9_	[M + H]^+^	595.3841	595.3864	3.86	595.3864 (72.38), 433.2563 (1.99), 397.3130 (3.88), 379.3032 (4.41), 301.2546 (19.14), 289.2170 (14.86), 283.2429 (100.00), 271.2065 (35.10), 253.1959 (16.57)	MS/MS
**42**	32.47	5β-Spirost-25(27)-en-2β,3β-diol 3-*O*-Xyl(1→3)-Gal ^#^	C_38_H_60_O_13_	[M + H]^+^	725.4107	725.4124	2.34	725.4124 (3.41), 707.4037 (1.07), 593.3687 (4.44), 431.3184 (39.43), 413.3065 (61.00), 395.2963 (23.55), 289.2172 (100.00), 283.2428 (6.71), 271.2066 (73.42), 253.1959 (33.59)	MS/MS
**43**	34.57	5β-Spirost-25(27)-en-3β-ol-12-one 3-*O*-Glc(1→3)-Glc ^#^	C_39_H_60_O_14_	[M + H]^+^	753.4056	753.4091	4.65	753.4091 (17.55), 591.3540 (15.97), 429.3008 (100.00), 411.2898 (30.12), 393.2813 (21.08), 299.2374 (32.01), 281.2271 (18.84)	MS/MS
**44**	35.94	25(*S*)-Schidigera-saponin F_1_ *	C_44_H_72_O_18_	[M + H]^+^	889.4791	889.4822	3.49	889.4822 (4.20), 757.4364 (1.66), 595.3850 (5.93), 433.3320 (100.00), 415.3212 (46.55), 397.3115 (9.38), 379.3004 (2.15), 301.2536 (2.40), 289.2171 (65.36), 283.2439 (3.56), 271.2063 (71.42), 253.1959 (34.71)	MS/MS, standard isolated by our lab
**45**	36.36	25(*R*)-Schidigera-saponin F_1_	C_44_H_72_O_18_	[M + H]^+^	889.4791	889.4821	3.37	889.4821 (4.20), 757.4364 (1.66), 595.3850 (5.93), 433.3327 (100.00), 415.3225 (67.62), 397.3102 (8.19), 379.3004 (2.15), 301.2536 (2.40), 289.2170 (62.53), 283.2441 (3.60), 271.2064 (63.65), 253.1961 (31.42)	MS/MS, literature [5]
**46**	36.77	Yucca spirostanoside C_2_ *	C_38_H_58_O_13_	[M + H]^+^	723.3950	723.3977	3.73	723.3977 (4.31), 705.3864 (12.41), 687.3758 (6.45), 591.3540 (12.47), 573.3441 (10.91), 555.3307 (1.61), 429.3013 (43.67), 411.2910 (54.19), 393.2802 (44.17), 375.2698 (19.47), 317.2483 (18.16), 299.2378 (100.00), 281.2270 (41.92), 269.1910 (9.61), 251.1804 (3.87)	MS/MS, standard reproted by literature [1]
**47**	37.37	25(*S*)-Schidigera-saponin F_2_ *	C_39_H_64_O_14_	[M + H]^+^	757.4369	757.4332	−4.88	757.4332 (1.82), 595.3813 (2.03), 433.3332 (26.30), 415.3218 (39.12), 397.3109 (9.34), 301.2543 (2.12), 289.2172 (100.00), 283.2439 (2.92), 271.2066 (62.23), 253.1962 (27.30)	MS/MS, standard isolated by our lab
**48**	37.47	(25*S*)-5β-Spirostan-3β-ol-12-one 3-*O*-Xyl(1→3)-Gal ^#^	C_38_H_60_O_13_	[M + H]^+^	725.4107	725.4122	2.07	725.4122 (14.77), 707.4008 (11.22), 593.3725 (19.80), 575.3577 (4.45), 431.3171 (100.00), 413.3061 (51.82), 395.2970 (25.63), 377.2819 (2.98), 299.2374 (34.86), 281.2279 (18.57), 269.1892 (2.30)	MS/MS
**49**	37.66	(25*R*)-5β-Spirostan-3β-ol-12-one 3-*O*-Xyl(1→3)-Gal ^#^	C_38_H_60_O_13_	[M + H]^+^	725.4107	725.4090	−2.34	725.4090 (8.00), 707.4029 (20.20), 593.3702 (19.76), 575.3591 (8.89), 431.3179 (100.00), 413.3059 (42.51), 395.2968 (38.05), 377.2858 (14.82), 299.2380 (62.72), 281.2271 (39.04), 269.1911 (7.12)	MS/MS
**50**	37.77	25(*R*/*S*)-Schidigera-saponin F	C_39_H_64_O_14_	[M + H]^+^	757.4369	757.4366	−0.40	757.4366 (1.23), 595.3821 (2.35), 433.3328 (27.43), 415.3219 (39.86), 397.3109 (9.03), 301.2545 (1.68), 289.2172 (100.00), 283.2439 (2.75), 271.2066 (58.68), 253.1962 (25.66)	MS/MS, literature [5]
**51**	37.78	(25*R*)-5β-spirostan-3β-ol-12-one 3-*O*-Glc(1→3)-Glc	C_39_H_62_O_14_	[M + H]^+^	755.4212	755.4230	2.38	755.4230 (9.78), 737.4004 (8.92), 593.3699 (40.52), 431.3174 (100.00), 413.3055 (52.49), 395.2940 (30.39), 377.2894 (6.82), 299.2377 (37.42), 281.2268 (28.82), 269.1898 (10.33)	MS/MS
**52**	38.27	Yucca spirostanoside C_1_ *	C_33_H_50_O_9_	[M + H]^+^	591.3528	591.3548	3.38	591.3548 (35.36), 573.3416 (55.79), 555.3334 (23.91), 429.3005 (6.86), 411.2913 (20.78), 393.2804 (36.45), 375.2680 (23.42), 317.2502 (5.55), 299.2381 (100.00), 281.2274 (47.22), 251.1799 (5.71)	MS/MS, standard reproted by literature [1]
**53**	38.38	25(*S*)-5β-Spirostan-2β,3β-diol 3-*O*-Glc(1→3)-Gal ^#^	C_39_H_64_O_14_	[M + H]^+^	757.4369	757.4352	−2.24	757.4352 (1.76), 595.3810 (4.74), 433.3313 (39.86), 415.3202 (62.66), 397.3110 (10.96), 301.2529 (4.71), 289.2158 (64.58), 283.2425 (4.70), 271.2055 (100.00), 253.1949 (33.87)	MS/MS
**54**	38.50	5β-Spirost-25(27)-en-1β,3β-diol 3-*O*-Xyl(1→3)-Glc ^#^	C_38_H_60_O_13_	[M + H]^+^	725.4107	725.4122	2.07	725.4122 (2.94), 707.4010 (5.95), 593.3736 (5.34), 431.3182 (25.61), 413.3066 (29.77), 395.2963 (44.26), 377.2851 (25.24), 289.2184 (1.32), 283.2433 (23.41), 271.2064 (27.92), 253.1958 (100.00)	MS/MS
**55**	38.74	25(*R*)-5β-Spirostan-2β,3β-diol 3-*O*-Glc(1→3)-Gal ^#^	C_39_H_64_O_14_	[M + H]^+^	757.4369	757.4352	−2.24	757.4352 (1.76), 595.3810 (4.74), 433.3313 (39.86), 415.3202 (62.66), 397.3110 (10.96), 301.2529 (4.71), 289.2158 (64.58), 283.2425 (4.70), 271.2055 (100.00), 253.1949 (33.87)	MS/MS
**56**	39.49	(25*S*)-Yucca spirostanoside E_2_ *	C_38_H_60_O_13_	[M + H]^+^	725.4107	725.4116	1.24	725.4116 (9.29), 707.4016 (23.00), 689.3932 (12.09), 593.3711 (26.56), 575.3602 (11.10), 557.3492 (3.64), 431.3180 (65.59), 413.3069 (48.22), 395.2965 (38.06), 377.2856 (14.46), 317.2494 (20.08), 299.2380 (100.00), 281.2272 (41.64), 269.1906 (4.80)	MS/MS, standard reproted by literature [6]
**57**	40.03	(25*R*)-Yucca spirostanoside E_2_ *	C_38_H_60_O_13_	[M + H]^+^	725.4107	725.4116	1.24	725.4116 (9.02), 707.4017 (22.76), 689.3934 (12.29), 593.3711 (26.66), 575.3603 (10.76), 557.3492 (3.70), 431.3180 (64.79), 413.3068 (48.23), 395.2965 (37.82), 377.2856 (14.26), 317.2494 (19.80), 299.2380 (100.00), 281.2272 (41.49), 269.1906 (4.40)	MS/MS, standard reproted by literature [6]
**58**	40.64	25(*S*)-5β-Spirostan-1β,3β-diol 3-*O*-Xyl(1→3)-Glc ^#^	C_38_H_62_O_13_	[M + H]^+^	727.4263	727.4291	3.85	727.4291 (2.08), 709.4142 (5.57), 691.4039 (28.21), 595.3818 (1.55), 577.3726 (3.78), 433.3325 (10.74), 415.3194 (20.52), 397.3095 (100.00), 379.2986 (39.97), 301.2521 (9.45), 289.2150 (6.16), 283.2416 (32.99), 271.2050 (28.80), 253.1946 (50.87)	MS/MS
**59**	40.83	25(*R*)-5β-Spirostan-1β,3β-diol 3-*O*-Xyl(1→3)-Glc ^#^	C_38_H_62_O_13_	[M + H]^+^	727.4263	727.4291	3.85	727.4291 (2.08), 709.4142 (5.57), 691.4039 (28.21), 595.3818 (1.55), 577.3726 (3.78), 433.3325 (10.74), 415.3194 (20.52), 397.3095 (100.00), 379.2986 (39.97), 301.2521 (9.45), 289.2150 (6.16), 283.2416 (32.99), 271.2050 (28.80), 253.1946 (50.87)	MS/MS
**60**	40.86	(25*S*)-Yucca spirostanoside E_1_ *	C_33_H_52_O_9_	[M + H]^+^	593.3684	593.3704	3.37	593.3704 (100.00), 575.3602 (37.61), 557.3490 (19.57), 431.3152 (2.97), 413.3061 (10.24), 395.2963 (24.86), 377.2859 (29.37), 317.2491 (11.16), 299.2374 (56.10), 281.2276 (36.70), 269.1902 (7.85)	MS/MS, standard reproted by literature [6]
**61**	40.90	(25*R*)-Yucca spirostanoside E_1_ *	C_33_H_52_O_9_	[M + H]^+^	593.3684	593.3701	2.86	593.3701 (100.00), 575.3604 (36.38), 557.3493 (17.61), 431.3148 (3.33), 413.3063 (10.21), 395.2965 (23.26), 377.2864 (11.69), 317.2491 (12.05), 299.2374 (53.58), 281.2278 (35.90), 269.1901 (8.53)	MS/MS, standard reproted by literature [6]
**62**	42.57	5β-Spirost-25(27)-en-3-*O*-Glc(1→3)-[Glc(1→2)]-Gal ^#^	C_45_H_72_O_18_	[M + H]^+^	901.4791	901.4750	−4.55	901.4750 (0.65), 739.4295 (0.80), 577.3744 (3.54), 415.3223 (25.12), 397.3108 (5.68), 379.2019 (0.59), 273.2218 (41.38), 255.2113 (100.00)	MS/MS
**63**	42.78	Rhodea sapogenin 3-*O*-Glc	C_33_H_54_O_9_	[M + H]^+^	595.3841	595.3860	3.19	595.3860 (10.71), 577.3752 (8.21), 559.3649 (39.26), 433.2632 (1.68), 415.3235 (3.08), 397.3110 (21.08), 379.3017 (23.52), 301.2536 (2.98), 289.2166 (0.32), 283.2427 (27.79), 271.2063 (19.19), 253.1957 (100.00)	MS/MS, literature [13]
**64**	42.81	Isorhodea sapogenin 3-*O*-Glc	C_33_H_54_O_9_	[M + H]^+^	595.3841	595.3867	4.37	595.3867 (10.56), 577.3759 (8.42), 559.3651 (39.58), 433.2600 (2.43), 415.3236 (2.96), 397.3111 (22.01), 379.3015 (26.31), 301.2530 (3.19), 289.2188 (0.07), 283.2429 (32.12), 271.2063 (21.55), 253.1958 (100.00)	MS/MS, literature [13]
**65**	43.60	Schidigera-saponin A_2_	C_44_H_70_O_17_	[M + H]^+^	871.4686	871.4683	−0.34	871.4683 (5.58), 739.4295 (0.44), 577.3759 (6.59), 415.3227 (30.22), 397.3117 (10.42), 379.3015 (1.04), 273.2224 (58.89), 255.2119 (100.00)	MS/MS, literature [5]
**66**	44.37	Schidigera-saponin A_3_ *	C_45_H_72_O_18_	[M + H]^+^	901.4791	901.4806	1.66	901.4806 (0.39), 739.4223 (0.60), 577.3760 (4.43), 415.3221 (19.02), 397.3107 (5.97), 273.2225 (43.29), 255.2117 (100.00)	MS/MS, standard reproted by literature [1]
**67**	45.70	5β-Spirost-25(27)-en-3-*O*-Api(1→3)-[Glc(1→2)]-Glc ^#^	C_44_H_70_O_17_	[M + H]^+^	871.4686	871.4683	−0.34	871.4683 (3.49), 739.4295 (0.26), 577.3759 (3.89), 415.3226 (20.38), 397.3119 (7.62), 379.3015 (0.85), 273.2224 (51.02), 255.2119 (100.00)	MS/MS
**68**	45.85	25(*S*)-Schidigera-saponin D_4_	C_45_H_74_O_18_	[M + H]^+^	903.4948	903.4932	−1.77	903.4932 (0.26), 741.4412 (6.04), 579.3887 (100.00), 417.3372 (39.49), 399.3286 (10.50), 273.2224 (37.44), 255.2121 (41.63)	MS/MS, literature [5]
**69**	46.25	5β-Spirost-25(27)-en-3β,12β-diol ^#^	C_27_H_42_O_4_	[M + H]^+^	431.3156	431.3175	4.41	431.3175 (77.53), 413.3070 (10.73), 395.2965 (11.49), 289.2169 (70.91), 283.2458 (100.00), 271.2064 (13.97), 253.1960 (11.87)	MS/MS
**70**	46.40	25(*R/S*)-5β-Spirostan-2β,3β-diol 3-*O*-Xyl(1→3)-Gal ^#^	C_38_H_62_O_13_	[M + H]^+^	727.4263	727.4297	4.67	727.4297 (2.08), 595.3818 (3.68), 577.3726 (3.81), 433.3324 (52.35), 415.3224 (98.24), 397.3120 (7.81), 301.2544 (10.27), 289.2170 (91.96), 271.2063 (100.00), 253.1959 (41.68)	MS/MS
**71**	46.55	Ys-IV	C_45_H_74_O_18_	[M + H]^+^	903.4948	903.4932	−1.77	903.4932 (0.26), 741.4412 (6.04), 579.3928 (3.99), 417.3380 (48.45), 399.3276 (6.30), 273.2220 (100.00), 255.2115 (51.54)	MS/MS, literature [5]
**72**	46.88	Schidigera-saponin A_1_ *	C_44_H_70_O_17_	[M + H]^+^	871.4686	871.4693	0.80	871.4693 (0.94), 739.4294 (0.07), 577.3754 (5.44), 415.3222 (22.22), 397.3116 (9.62), 379.3011 (0.29), 273.2223 (43.60), 255.2118 (100.00)	MS/MS, standard reproted by literature [1]
**73**	47.26	25(*S*)-Schidegera-saponin D_2_	C_44_H_72_O_17_	[M + H]^+^	873.4842	873.4800	−4.81	873.4800 (0.32), 741.4323 (0.61), 579.3882 (7.97), 417.3358 (35.70), 399.3250 (15.86), 381.3137 (2.33), 273.2207 (35.79), 255.2104 (100.00)	MS/MS, literature [5]
**74**	48.01	25(*R*)-Schidegera-saponin D_2_	C_44_H_72_O_17_	[M + H]^+^	873.4842	873.4879	4.24	873.4879 (0.32), 741.4323 (0.61), 579.3882 (7.97), 417.3358 (35.70), 399.3250 (15.86), 381.3137 (2.33), 273.2207 (35.79), 255.2104 (100.00)	MS/MS, literature [5]
**75**	48.22	25(*S*)-Schidigera-saponin D_3_ *	C_45_H_74_O_18_	[M + H]^+^	903.4948	903.4937	−1.22	903.4937 (2.00), 772.1501 (1.73), 579.3891 (3.38), 417.3375 (25.17), 399.3263 (7.83), 273.2226 (27.94), 255.2118 (100.00)	MS/MS, standard isolated by our lab
**76**	48.49	Mexogenin	C_27_H_42_O_5_	[M + H]^+^	447.3105	447.3113	1.79	447.3113 (32.13), 429.3023 (100.00), 411.2913 (19.45), 315.2332 (9.37). 285.1844 (1.12), 253.1693 (1.93)	MS/MS, literature [5]
**77**	48.82	Timosaponin A III	C_39_H_64_O_13_	[M + H]^+^	741.4420	741.4452	4.32	741.4452 (2.37), 579.3932 (3.64), 417.3382 (10.01), 399.3280 (9.12), 285.2587 (3.32), 273.2222 (39.02), 255.2117 (100.00)	MS/MS, literature [14]
**78**	48.99	Ys-III	C_45_H_74_O_18_	[M + H]^+^	903.4948	903.4937	−1.22	903.4937 (2.00), 772.1495 (0.91), 579.3891 (3.59), 417.3371 (27.58), 399.3271 (9.06), 273.2222 (30.82), 255.2118 (100.00)	MS/MS, literature [5]
**79**	49.49	Asparinin A	C_39_H_64_O_13_	[M + H]^+^	741.4420	741.4432	1.62	741.4432 (1.50), 579.3926 (4.93), 417.3386 (10.53), 399.3273 (8.40), 285.2581 (3.77), 273.2222 (39.90), 255.2118 (100.00)	MS/MS, literature [15]
**80**	50.00	Tuberosine A	C_33_H_54_O_9_	[M + H]^+^	595.3841	595.3868	4.53	595.3868 (24.29), 433.3332 (23.11), 415.3225 (56.28), 397.3112 (17.73), 301.2546 (6.96), 289.2172 (75.61), 283.2436 (10.33), 271.2067 (100.00), 253.1959 (52.17)	MS/MS, literature [16]
**81**	50.00	(25*R*)-5β-Spirostan-2β,3β-diol 3-*O*-Glc ^#^	C_33_H_54_O_9_	[M + H]^+^	595.3841	595.3868	4.53	595.3868 (23.62), 433.3333 (21.65), 415.3224 (60.96), 397.3110 (17.71), 301.2522 (5.47), 289.2172 (73.47), 283.2436 (8.85), 271.2067 (100.00), 253.1959 (52.88)	MS/MS
**82**	50.01	5β-Spirost-25(27)-en-3β-yl 3-*O*-Glc(1→2)-Glc ^#^	C_39_H_62_O_13_	[M + H]^+^	739.4263	739.4282	2.57	739.4282 (1.09), 577.3763 (2.43), 415.3220 (4.80), 397.3116 (7.67), 379.3027 (1.41), 285.2587 (2.73), 273.2221 (30.31), 255.2116 (100.00)	MS/MS
**83**	50.76	25(*S*)-Schidigera-saponin D_5_ *	C_39_H_64_O_13_	[M + H]^+^	741.4420	741.4446	3.51	741.4446 (4.25), 579.3954 (0.74), 417.3398 (6.71), 399.3265 (4.94), 285.2582 (1.86), 273.2222 (29.87), 255.2118(100.00)	MS/MS, standard reproted by literature [6]
**84**	51.46	25(*S*)-Schidegera-saponin D_1_ *	C_44_H_72_O_17_	[M + H]^+^	873.4842	873.4876	3.89	873.4876 (0.32), 741.4323 (0.61), 579.3897 (6.82), 417.3355 (35.08), 399.3251 (14.24), 381.3162 (2.60), 273.2209 (32.22), 255.2104 (100.00)	MS/MS, standard reproted by literature [6]
**85**	51.65	25(*R*)-Schidigera-saponin D_5_ *	C_39_H_64_O_13_	[M + H]^+^	741.4420	741.4434	1.89	741.4434 (2.52), 579.3890 (0.90), 417.3380 (7.65), 399.3269 (5.68), 285.2586 (2.06), 273.2211 (30.08), 255.2117 (100.00)	MS/MS, standard reproted by literature [6]
**86**	52.10	25(*R*)-Schidegera-saponin D_1_ *	C_44_H_72_O_17_	[M + H]^+^	873.4842	873.4876	3.89	873.4876 (0.32),741.4323 (0.61), 579.3897 (6.82), 417.3355 (35.08), 399.3251 (14.24), 381.3162 (2.60), 273.2209 (32.22), 255.2104 (100.00)	MS/MS, standard reproted by literature [6]
**87**	53.62	Schidegeragenin B	C_27_H_40_O_3_	[M + H]^+^	429.2999	429.3015	3.73	429.3015 (100.00), 411.2890 (43.35), 393.2812 (36.09), 299.2374 (19.35), 281.2256 (19.25), 269.1908 (8.16)	MS/MS, literature [5]
**88**	53.66	(25*S*)-5β-Spirostan-3β-yl 3-*O*-Glc(1→3)-Gal ^#^	C_39_H_64_O_13_	[M + H]^+^	741.4420	741.4442	2.97	741.4442 (0.53), 417.3377 (3.74), 399.3259 (2.41), 273.2216 (34.83), 255.2114 (100.00)	MS/MS
**89**	53.99	(25*R/S*)-12β-Hydroxysmilagenin	C_27_H_44_O_4_	[M + H]^+^	433.3312	433.3321	2.08	433.3321 (100.00), 289.2173 (21.95), 283.2428 (22.32), 271.2060 (9.32), 253.1963 (10.37)	MS/MS, literature [10]
**90**	54.20	5β-Spirost-25(27)-en-3β-ol 3-*O*-Xyl(1→3)-Gal ^#^	C_38_H_60_O_12_	[M + H]^+^	709.4158	709.4167	1.27	709.4167 (12.74), 567.3175 (1.32), 435.2757 (1.81),415.3192 (6.10), 397.3103 (4.17), 273.2222 (23.06), 255.2116 (100.00)	MS/MS
**91**	54.31	(25*R*)-5β-Spirostan-3β-yl 3-*O*-Glc(1→3)-Gal ^#^	C_39_H_64_O_13_	[M + H]^+^	741.4420	741.4452	4.32	741.4452 (2.12), 579.3917 (1.41), 417.3338 (8.83), 399.3278 (7.04), 285.2578 (2.34), 273.2221 (31.71), 255.2117 (100.00)	MS/MS
**92**	55.22	5β-Spirost-25(27)-en-3β-yl 3-*O*-Glc(1→3)-Glc ^#^	C_39_H_62_O_13_	[M + H]^+^	739.4263	739.4254	−1.22	739.4254 (7.51), 577.3753 (1.71), 415.3225 (6.43), 397.3108 (8.13), 379.3026 (1.03), 285.2591 (2.67), 273.2221 (33.28), 255.2117 (100.00)	MS/MS
**93**	55.22	25(*R/S*)-5β-Spirostan-3β-ol 3-*O*-Gal	C_33_H_54_O_8_	[M + H]^+^	579.3891	579.3913	3.80	579.3913 (14.18), 435.2762 (11.61), 399.3256 (0.97), 285.2598 (0.93), 273.2221 (14.70), 255.2116 (100.00)	MS/MS, literature [17]
**94**	55.32	25(*S*)-5β-Spirostan-3β-ol 3-*O*-Xyl(1→3)-Gal ^#^	C_38_H_62_O_12_	[M + H]^+^	711.4314	711.4335	2.95	711.4335 (7.88), 567.3163 (2.22), 435.2789 (2.10), 417.3373 (6.00), 399.3256 (8.41), 285.2584 (4.27), 273.2220 (29.40), 255.2114 (100.00)	MS/MS
**95**	55.51	(25*S*)-5β-Spirostan-3β-ol-12-one	C_27_H_42_O_4_	[M + H]^+^	431.3156	431.3177	4.87	431.3177 (100.00), 413.3067(46.79), 395.2965 (25.11), 377.2840(0.25), 317.2492(9.30), 299.2379 (39.06), 281.2274 (24.81)	MS/MS, literature [5]
**96**	56.04	25(*R*)-5β-Spirostan-3β-ol 3-*O*-Xyl(1→3)-Gal ^#^	C_38_H_62_O_12_	[M + H]^+^	711.4314	711.4305	−1.27	711.4305 (6.51), 579.3914 (1.57), 417.3357 (3.88), 399.3264 (4.26), 285.2581 (2.83), 273.2220 (26.63), 255.2116 (100.00)	MS/MS
**97**	56.37	Yucca spirostanoside A_2_ *	C_38_H_60_O_12_	[M + H]^+^	709.4158	709.4163	0.70	709.4163 (4.36), 577.3773 (1.06), 415.3223 (3.53), 397.3117 (4.25), 379.3006 (0.11), 285.2590 (1.71), 273.2219 (26.74), 255.2116 (100.00)	MS/MS, standard reproted by literature [1]
**98**	56.43	25(*S*)-5β-Spirostan-3β-ol 3-*O*-Glc(1→3)-Glc ^#^	C_39_H_64_O_13_	[M + H]^+^	741.4420	741.4414	–0.81	741.4414 (1.19), 579.3919 (3.14), 417.3380 (11.15), 399.3275 (14.43), 285.2584 (4.44), 273.2223 (43.87), 255.2118 (100.00)	MS/MS
**99**	57.03	25(*R*)-5β-spirostan-3β-ol 3-*O*-Glc(1→3)-Glc ^#^	C_39_H_64_O_13_	[M + H]^+^	741.4420	741.4436	2.16	741.4436 (1.11), 579.3913 (3.32), 417.3377 (11.40), 399.3276 (14.08), 285.2584 (4.09), 273.2223 (40.84), 285.2579 (1.15), 255.2118 (100.00)	MS/MS
**100**	57.13	Yucca spirostanoside A_1_ *	C_33_H_52_O_8_	[M + H]^+^	577.3735	577.3737	0.35	577.3737 (10.11), 415.3222 (20.05), 397.3143 (0.89), 379.3027 (0.60), 273.2222 (17.54), 255.2117 (100.00)	MS/MS, standard reproted by literature [1]
**101**	57.79	(25*R*)-Spirostan-3β-ol-12-one	C_27_H_42_O_4_	[M + H]^+^	431.3156	431.3162	1.39	431.3162 (100.00), 413.3064 (37.68), 395.2967 (23.34), 377.2850 (5.94), 317.2499 (2.84), 299.2378 (25.93), 281.2275 (23.20)	MS/MS, literature [5]
**102**	57.99	25(*S*)-5β-Spirostan-3β-ol 3-*O*-Xyl(1→3)-Glc ^#^	C_38_H_62_O_12_	[M + H]^+^	711.4314	711.4305	−1.27	711.4305 (5.95), 579.3902 (1.23), 417.3376 (7.78), 399.3273 (9.77), 285.2580 (3.40), 273.2221 (33.53), 255.2115 (100.00)	MS/MS
**103**	58.02	25(*R*)-5β-Spirostan-3β-ol 3-*O*-Xyl(1→3)-Glc ^#^	C_38_H_62_O_12_	[M + H]^+^	711.4314	711.4323	1.27	711.4323 (1.77), 579.3886 (0.76), 417.3372 (6.68), 399.3280 (8.46), 285.2591 (3.66), 273.2224 (33.80), 255.2119 (100.00)	MS/MS
**104**	58.57	(25*R*)-5β-Spirostan-3β-ol 3-*O*-*G*lc *	C_33_H_54_O_8_	[M + H]^+^	579.3891	579.3918	4.66	579.3918 (16.27), 435.2755 (11.79), 417.3367 (0.02), 399.3271 (1.87), 381.3169 (0.03), 285.2588 (1.11), 273.2220 (17.83), 255.2118 (100.00)	MS/MS, standard reproted by literature [6]
**105**	58.62	Asparagoside A *	C_33_H_54_O_8_	[M + H]^+^	579.3891	579.3911	3.45	579.3911 (16.39), 435.2753 (12.48), 417.3367 (0.02), 399.3271 (1.79), 381.3169(0.02), 285.2588(1.15), 273.2220 (17.92), 255.2117 (100.00)	MS/MS, standard reproted by literature [6]
**106**	58.72	Samogenin/Markogenin	C_27_H_44_O_4_	[M + H]^+^	433.3312	433.3330	4.15	433.3330 (53.36), 415.3216 (5.59), 301.2562 (1.14), 289.2174 (100.00), 283.2453 (1.21), 271.2068 (56.56), 253.1962 (33.89)	MS/MS, literature [5]
**107**	60.36	(25*R/S*)-Isorhodeasapogenin	C_27_H_44_O_4_	[M + H]^+^	433.3312	433.3324	2.77	433.3324 (19.22), 415.3218 (10.46), 301.2530 (4.85), 289.2169 (5.66), 283.2431 (17.61), 271.2064 (17.78), 253.1961 (28.29)	MS/MS, literature [13]
**108**	61.29	Schidigeragenin A_1_	C_27_H_42_O_3_	[M + H]^+^	415.3207	415.3225	4.33	415.3225 (85.22), 397.3115 (18.74), 379.3015 (8.75), 285.2586(2.26), 273.2224 (100.00), 255.2119 (86.40)	MS/MS, literature [5]
**109**	63.73	Sarsasapogenin	C_27_H_44_O_3_	[M + H]^+^	417.3363	417.3382	4.55	417.3382 (35.52), 285.2589 (1.44), 273.2226 (100.00), 255.2120 (70.59)	MS/MS, literature [5]
**110**	64.08	Smilagenin	C_27_H_44_O_3_	[M + H]^+^	417.3363	417.3348	−3.59	417.3348 (35.68), 285.2599 (1.38), 273.2226 (100.00), 255.2117 (72.83)	MS/MS, literature [5]

* The compounds unambiguously identified with the reference standards comparison; ^#^ The potential new compounds; The underline annotation indicated that the structures of compounds were proved by targeted isolation.

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
