# Peer review of "Identification and Structural Analysis of Spirostanol Saponin from Yucca schidigera by Integrating Silica Gel Column Chromatography and Liquid Chromatography/Mass Spectrometry Analysis"

_molecules, 2020, doi:10.3390/molecules25173848_

Round 1

Reviewer 1 Report

The manuscript by J. Ruam et al. dedicated to the identification of saponins from Yucca plant represents an interesting and important study due to high biological activity of the target class of compounds and advanced analytical techniques used in this work. The data on mass spectrometry and NMR identification are of good quality and beyond doubt. However, there are some issues that do not allow recommending the manuscript for publication in its present form and which must be clarified. The manuscript should be seriously rewritten before further consideration.

The main concern is the focus of the study. In my opinion, this is an identification of saponins using the sophisticated but common methodology. I do not agree with the authors’ claim that orthogonal chromatography – high resolution mass spectrometry method was developed. The method used is the preliminary column fractionation of plant extract very common in phytochemistry with further HPLC-MS/MS analysis of the obtained fractions and cannot be considered as OC-MS/MS.

Introduction section. The introduction looks too scanty. The aim of this work was not rationalized and remained unclear. If the authors aim to study the chemical composition of a plant, then attention should be drawn to the information already available in the literature on this issue. If the development and application of the method of orthogonal chromatography is at the forefront, then it is necessary to more clearly describe the proposed technology and its advantages in the Experimental part or in the Results and discussion section.

Table 1. Are error values measured in ppm? In some cases they are confusing. Row 36: Theoretical mass 727.4263, measured 727.4308, Delta must be more than 6 ppm, not 0.3! This value is above accuracy threshold (5 ppm) mentioned in experimental part (section S4).

Experimental part is very important in analytical chemistry article. Thus, I would recommend to include it in the article removing from supplementary.  

Supplementary S1. “Positive ion mode ESI Q Orbitrap MS were obtained on a Thermo Q Orbitrap MS spectrometer (Agilent Corp., Santa Clara, CA, USA)”. What particular model of mass spectrometer was used? Q Exactive? LTQ Orbitrap? Why MS manufacturer is Agilent? What mass resolution was used?

Supplementary S4. “Ultra-high purity helium nitrogen (N2) and high purity nitrogen (N2) were used as the collision gas and the sheath/auxiliary gas, respectively”. Helium or nitrogen?

Experimental part and supplementary. Column fractionation procedure was not described properly. What columns were used (ID, length, material, packing. particle size, flow rates, fractions volumes and their collection etc.).

Conclusion section declares that “comprehensive and rapid characterization method for spirostanol saponins was set up for the first time”. What is rapidity of the proposed method?

Reviewer 2 Report

The paper: Identification and structural analysis of spirostanol saponin from Yucca schidigera by integrating silica gel column chromatography and liquid chromatography/mass spectrometry analysis is of a high scientific level, well structured, the working methodology and the results obtained are presented in detail.  

 I recommend, however, to improve the literature study from introduction with previous data regarding the identification and quantification of saponins from Yucca schidigera.

Round 2

Reviewer 1 Report

The authors took my comments into account and significantly improved the quality of the manuscript. This is a good work in phytochemistry. I recommend accepting it for publication in its present form.

Author Response

Thank you for your previous comments, this manuscript has been revised carefully again for the grammatical errors and English language expression as well as the formats.